# An Exponential Lower Bound for Linearly-Realizable MDPs with Constant Suboptimality Gap

**Yuanhao Wang**
Princeton University
yuanhao@princeton.edu

**Ruosong Wang**
Carnegie Mellon University
ruosongw@andrew.cmu.edu

**Sham M. Kakade**
University of Washington and Microsoft Research
sham@cs.washington.edu

## Abstract

A fundamental question in the theory of reinforcement learning is: suppose the optimal $Q$-function lies in the linear span of a given $d$ dimensional feature mapping, is sample-efficient reinforcement learning (RL) possible? The recent and remarkable result of Weisz et al. (2020) resolves this question in the negative, providing an exponential (in $d$) sample size lower bound, which holds even if the agent has access to a generative model of the environment. One may hope that such a lower can be circumvented with an even stronger assumption that there is a *constant gap* between the optimal $Q$-value of the best action and that of the second-best action (for all states); indeed, the construction in Weisz et al. (2020) relies on having an exponentially small gap. This work resolves this subsequent question, showing that an exponential sample complexity lower bound still holds even if a constant gap is assumed. Perhaps surprisingly, this result implies an exponential separation between the online RL setting and the generative model setting, where sample-efficient RL is in fact possible in the latter setting with a constant gap. Complementing our negative hardness result, we give two positive results showing that provably sample-efficient RL is possible either under an additional low-variance assumption or under a novel hypercontractivity assumption.

## 1 Introduction

There has been substantial recent theoretical interest in understanding the means by which we can avoid the curse of dimensionality and obtain sample-efficient reinforcement learning (RL) methods [Wen and Van Roy, 2017, Du et al., 2019b,a, Wang et al., 2019, Yang and Wang, 2019, Lattimore et al., 2020, Yang and Wang, 2020, Jin et al., 2020, Cai et al., 2020, Zanette et al., 2020, Weisz et al., 2020, Du et al., 2020, Zhou et al., 2020b,a, Modi et al., 2020, Jia et al., 2020, Ayoub et al., 2020]. Here, the extant body of literature largely focuses on sufficient conditions for efficient reinforcement learning. Our understanding of what are the necessary conditions for efficient reinforcement learning is far more limited. With regards to the latter, arguably, the most natural assumption is linear realizability: we assume that the optimal $Q$-function lies in the linear span of a given feature map. The goal is to the obtain polynomial sample complexity under this linear realizability assumption alone.

This "linear $Q^*$ problem" was a major open problem (see Du et al. [2019a] for discussion), and a recent hardness result by Weisz et al. [2020] provides a negative answer. In particular, the result shows that even with access to a generative model, any algorithm requires an exponential number

35th Conference on Neural Information Processing Systems (NeurIPS 2021).

| Minimum Gap? | Generative Model | Online RL |
|:---:|:---:|:---:|
| No | Exponential [Weisz et al., 2020] | Exponential [Weisz et al., 2020] |
| Yes | Polynomial [Du et al., 2019a] | **Exponential (This work, Theorem 1)** |

Table 1: Known sample complexity results for RL with linear function approximation under realizability. "Exponential" refers to exponential lower bound (in the dimension or horizon), while "polynomial" refers to a polynomial upper bound.

of samples (in the dimension $d$ of the feature mapping) to find a near-optimal policy, provided the action space has exponential size.

With this question resolved, one may naturally ask what is the source of hardness for the construction in Weisz et al. [2020] and if there are additional assumptions that can serve to bypass the underlying source of this hardness. Here, arguably, it is most natural to further examine the suboptimality gap in the problem, which is the gap between the optimal $Q$-value of the best action and that of the second-best action; the construction in Weisz et al. [2020] does in fact fundamentally rely on having an exponentially small gap. Instead, if we assume the gap is lower bounded by a constant for all states, we may hope that the problem becomes substantially easier since with a finite number of samples (appropriately obtained), we can identify the optimal policy itself (i.e., the gap assumption allows us to translate value-based accuracy to the identification of the optimal policy itself). In fact, this intuition is correct in the following sense: with a generative model, it is not difficult to see that polynomial sample complexity is possible under the linear realizability assumption plus the suboptimality gap assumption, since the suboptimality gap assumption allows us to easily identify an optimal action for all states, thus making the problem tractable (see Section C in Du et al. [2019a] for a formal argument).

More generally, the suboptimality gap assumption is widely discussed in the bandit literature [Dani et al., 2008, Audibert and Bubeck, 2010, Abbasi-Yadkori et al., 2011] and the reinforcement learning literature [Simchowitz and Jamieson, 2019, Yang et al., 2020] to obtain fine-grained sample complexity upper bounds. More specifically, under the realizability assumption and the suboptimality gap assumption, it has been shown that polynomial sample complexity is possible if the transition is nearly deterministic [Du et al., 2019b, 2020] (also see Wen and Van Roy [2017]). However, it remains unclear whether the suboptimality gap assumption is sufficient to bypass the hardness result in Weisz et al. [2020], or the same exponential lower bound still holds even under the suboptimality gap assumption, when the transition could be stochastic and the generative model is unavailable. For the construction in Weisz et al. [2020], at the final stage, the gap between the value of the optimal action and its non-optimal counterparts will be exponentially small, and therefore the same construction does not imply an exponential sample complexity lower bound under the suboptimality gap assumption.

**Our contributions.** In this work, we significantly strengthen the hardness result in Weisz et al. [2020]. In particular, we show that in the online RL setting (where a generative model is unavailable) with exponential-sized action space, the exponential sample complexity lower bound still holds even under the suboptimality gap assumption. Complementing our hardness result, we show that under the realizability assumption and the suboptimality gap assumption, our hardness result can be bypassed if one further assumes the low variance assumption in Du et al. [2019b] [1], or a hypercontractivity assumption. Hypercontractive distributions include Gaussian distributions (with arbitrary covariance matrices), uniform distributions over hypercubes and strongly log-concave distributions [Kothari and Steinhardt, 2017]. This condition has been shown powerful for outlier-robust linear regression [Kothari and Steurer, 2017], but has not yet been introduced for reinforcement learning with linear function approximation.

Our results have several interesting implications, which we discuss in detail in Section 6. Most notably, our results imply an *exponential* separation between the standard reinforcement learning setting and the generative model setting. Moreover, our construction enjoys greater simplicity, making it more suitable to be generalized for other RL problems or to be presented for pedagogical purposes.

---

[1]We note that the sample complexity of the algorithm in Du et al. [2019b] has at least linear dependency on the number of actions, which is not sufficient for bypassing our hardness results which assumes an exponential-sized action space.

## 2   Related work

**Previous hardness results.**   Existing exponential lower bounds in RL [Krishnamurthy et al., 2016, Chen and Jiang, 2019] usually construct unstructured MDPs with an exponentially large state space. Du et al. [2019a] prove that under the approximate version of the realizability assumption, i.e., the optimal $Q$-function lies in the linear span of a given feature mapping approximately, any algorithm requires an exponential number of samples to find a near-optimal policy. The main idea in Du et al. [2019a] is to use the Johnson-Lindenstrauss lemma [Johnson and Lindenstrauss, 1984] to construct a large set of near-orthogonal feature vectors. Such idea is later generalized to other settings, including those in Wang et al. [2020a], Kumar et al. [2020], Van Roy and Dong [2019], Lattimore et al. [2020]. Whether the exponential lower bound still holds under the exact version of the realizability assumption is left as an open problem in Du et al. [2019a].

The above open problem is recently solved by Weisz et al. [2020]. They show that under the exact version of the realizability assumption, any algorithm requires an exponential number of samples to find a near-optimal policy assuming an exponential-sized action space. The construction in Weisz et al. [2020] also uses the Johnson-Lindenstrauss lemma to construct a large set of near-orthogonal feature vectors, with additional subtleties to ensure exact realizability.

Very recently, under the exact realizability assumption, strong lower bounds are proved in the offline setting [Wang et al., 2020b, Zanette, 2020, Amortila et al., 2020]. These work focus on the offline RL setting, where a fixed data distribution with sufficient coverage is given and the agent cannot interact with the environment in an online manner. Instead, we focus on the online RL setting in this paper.

**Existing upper bounds.**   For RL with linear function approximation, most existing upper bounds require representation conditions stronger than realizability. For example, the algorithms in Yang and Wang [2019, 2020], Jin et al. [2020], Cai et al. [2020], Zhou et al. [2020b,a], Modi et al. [2020], Jia et al. [2020], Ayoub et al. [2020] assume that the transition model lies in the linear span of a given feature mapping, and the algorithms in Wang et al. [2019], Lattimore et al. [2020], Zanette et al. [2020] assume completeness properties of the given feature mapping. In the remaining part of this section, we mostly focus on previous upper bounds that require only realizability as the representation condition.

For deterministic systems, under the realizability assumption, Wen and Van Roy [2017] provide an algorithm that achieves polynomial sample complexity. Later, under the realizability assumption and the suboptimality gap assumption, polynomial sample complexity upper bounds are shown if the transition is deterministic [Du et al., 2020], a generative model is available [Du et al., 2019a], or a low-variance condition holds [Du et al., 2019b]. Compared to the original algorithm in Du et al. [2019b], our modified algorithm in Section 5 works under a similar low-variance condition. However, the sample complexity in Du et al. [2019b] has at least linear dependency on the number of actions, whereas our sample complexity in Section 5 has no dependency on the size of the action space. Finally, Shariff and Szepesvári [2020] obtain a polynomial upper bound under the realizability assumption when the features for all state-action pairs are inside the convex hull of a polynomial-sized coreset and the generative model is available to the agent.

## 3   Preliminaries

### 3.1   Markov decision process (MDP) and reinforcement learning

An MDP is specified by $(\mathcal{S}, \mathcal{A}, H, P, \{R_h\}_{h \in [H]})$, where $\mathcal{S}$ is the state space, $\mathcal{A}$ is the action space with $|\mathcal{A}| = A$, $H \in \mathbb{Z}^+$ is the planning horizon, $P : \mathcal{S} \times \mathcal{A} \to \Delta_{\mathcal{S}}$ is the transition function and $R_h : \mathcal{S} \times \mathcal{A} \to \Delta_{\mathbb{R}}$ is the reward distribution. Throughout the paper, we occasionally abuse notation and use a scalar $a$ to denote the single-point distribution at $a$.

A (stochastic) policy takes the form $\pi = \{\pi_h\}_{h \in [H]}$, where each $\pi_h : \mathcal{S} \to \Delta_{\mathcal{A}}$ assigns a distribution over actions for each state. We assume that the initial state is drawn from a fixed distribution, i.e. $s_1 \sim \mu$. Starting from the initial state, a policy $\pi$ induces a random trajectory $s_1, a_1, r_1, \cdots, s_H, a_H, r_H$ via the process $a_h \sim \pi_h(\cdot)$, $r_h \sim R(\cdot | s_h, a_h)$ and $s_{h+1} \sim P(\cdot | s_h, a_h)$. For a policy $\pi$, denote the distribution of $s_h$ in its induced trajectory by $\mathcal{D}_h^\pi$.

Given a policy $\pi$, the $Q$-function (action-value function) is defined as

$$Q_h^\pi(s, a) := \mathbb{E}\left[\sum_{h'=h}^{H} r_{h'} | s_h = s, a_h = a, \pi\right],$$

while $V_h^\pi(s) := \mathbb{E}_{a \sim \pi_h(s)}[Q_h^\pi(s, a)]$. We denote the optimal policy by $\pi^*$, and the associated optimal $Q$-function and value function by $Q^*$ and $V^*$ respectively. Note that $Q^*$ and $V^*$ can also be defined via the Bellman optimality equation[2]:

$$V_h^*(s) = \max_{a \in \mathcal{A}} Q_h^*(s, a),$$
$$Q_h^*(s, a) = \mathbb{E}\left[R_h(s, a) + V_{h+1}^*(s_{h+1}) | s_h = s, a_h = a\right].$$

**The online RL setting.** In this paper, we aim to prove lower bound and upper bound in the online RL setting. In this setting, in each episode, the agent interacts with the unknown environment using a policy and observes rewards and the next states. We remark that the hardness result by Weisz et al. [2020] operates in the setting where a generative model is available to the agent so that the agent can transit to any state. Also, it is known that with a generative model, under the linear realizability assumption plus the suboptimality gap assumption, one can find a near-optimal policy with polynomial number of samples (see Section C in Du et al. [2019a] for a formal argument).

### 3.2 Linear $Q^\star$ function approximation

When the state space is large or infinite, structures on the state space are necessary for efficient reinforcement learning. In this work we consider linear function approximation. Specifically, there exists a feature map $\phi : \mathcal{S} \times \mathcal{A} \to \mathbb{R}^d$, and we will use linear functions of $\phi$ to represent $Q$-functions of the MDP. To ensure that such function approximation is viable, we assume that the optimal $Q$-function is realizable.

**Assumption 1** (Realizability). For all $h \in [H]$, there exists $\theta_h^* \in \mathbb{R}^d$ such that for all $(s, a) \in \mathcal{S} \times \mathcal{A}$, $Q_h^*(s, a) = \phi(s, a)^\top \theta_h^*$.

This assumption is widely used in existing reinforcement learning and contextual bandit literature [Du et al., 2019b, Foster and Rakhlin, 2020]. However, even for linear function approximation, realizability alone is not sufficient for sample-efficient reinforcement learning [Weisz et al., 2020]. In this work, we also impose the regularity condition that $\|\theta_h^*\|_2 = O(1)$ and $\|\phi(s, a)\|_2 = O(1)$, which can always be achieved via rescaling.

Another assumption that we will use is that the minimum suboptimality gap is lower bounded. As mentioned in the introduction, this assumption is common in bandit and reinforcement learning literature.

**Assumption 2** (Minimum Gap). For any state $s \in \mathcal{S}$, $a \in \mathcal{A}$, the suboptimality gap is defined as $\Delta_h(s, a) := V_h^*(s) - Q_h^*(s, a)$. We assume that $\min_{h \in [H], s \in \mathcal{S}, a \in \mathcal{A}} \{\Delta_h(s, a) : \Delta_h(s, a) > 0\} \geq \Delta_{\min}$.

## 4 Hard Instance with Constant Suboptimality Gap

We now present our main hardness result:

**Theorem 1.** *Consider an arbitrary online RL algorithm that takes the feature mapping $\phi : \mathcal{S} \times \mathcal{A} \to \mathbb{R}^d$ as input. In the online RL setting, there exists an MDP with a feature mapping $\phi$ satisfying Assumption 1 and Assumption 2 with $\Delta_{\min} = \Omega(1)$, such that the algorithm requires $\min\{2^{\Omega(d)}, 2^{\Omega(H)}\}$ samples to find a policy $\pi$ with*

$$\mathbb{E}_{s_1 \sim \mu} V^\pi(s_1) \geq \mathbb{E}_{s_1 \sim \mu} V^*(s_1) - 0.05$$

*with probability $0.1$.*

The remainder of this section provides the construction of a hard family of MDPs where $Q^*$ is linearly realizable and has constant suboptimality gap and where it takes exponential samples to learn a

---

[2]We additionally define $V_{H+1}(s) = 0$ for all $s \in \mathcal{S}$.

near-optimal policy. Each of these hard MDPs can roughly be seen as a "leaking complete graph" (see detailed transtion probabilities below). Information about the optimal policy can only be gained by: (1) taking the optimal action; (2) reaching a non-terminal state at level $H$. We will show that when there are exponentially many actions, both events happen with negligible probability unless exponentially many trajectories are played.

## 4.1 Construction of the MDP family

In this section we describe the construction of the hard instance (the hard MDP family) in detail. Let $m$ be an integer to be determined. The state space is $\{\bar{1}, \cdots, \bar{m}, f\}$. The special state $f$ is called the *terminal state*. At state $\bar{i}$, the set of available actions is $[m] \setminus \{i\}$; at the terminal state $f$, the set of available actions is $[m-1]$. [3] In other words there are $m-1$ actions available at each state. Each MDP in this family is specified by an index $a^* \in [m]$ and denoted by $\mathcal{M}_{a^*}$. In other words, there are $m$ MDPs in this family.

In order to construct the MDP family, we first find a set of approximately orthogonal vectors by leveraging the Johnson-Lindenstrauss lemma [Johnson and Lindenstrauss, 1984].

**Lemma 1** (Johnson-Lindenstrauss). *For any $\gamma > 0$, if $m \leq \exp(\frac{1}{8}\gamma^2 d')$, there exists $m$ unit vectors $\{v_1, \cdots, v_m\}$ in $\mathbb{R}^{d'}$ such that for all $i, j \in [m]$ such that $i \neq j$, $|\langle v_i, v_j \rangle| \leq \gamma$.*

We will set $\gamma = \frac{1}{4}$ and $m = \lfloor \exp(\frac{1}{8}\gamma^2 d) \rfloor$. By Lemma 1, we can find such a set of $d$-dimensional unit vectors $\{v_1, \cdots, v_m\}$. For the clarity of presentation, we will use $v_i$ and $v(i)$ interchangeably. The construction of $\mathcal{M}_{a^*}$ is specified below.

**Features.** The feature map, which maps state-action pairs to $d$ dimensional vectors, is defined as

$$\phi(\overline{a_1}, a_2) := \left( \left\langle v(a_1), v(a_2) \right\rangle + 2\gamma \right) \cdot v(a_2), \qquad \forall a_1, a_2 \in [m], a_1 \neq a_2,$$
$$\phi(f, \cdot) := \mathbf{0}.$$

Note that the feature map is independent of $a^*$ and is shared across the MDP family.

**Rewards.** For $1 \leq h < H$, the rewards are defined as

$$R_h(\overline{a_1}, a^*) := \left\langle v(a_1), v(a^*) \right\rangle + 2\gamma,$$
$$R_h(\overline{a_1}, a_2) := -2\gamma \left[ \left\langle v(a_1), v(a_2) \right\rangle + 2\gamma \right], \qquad (a_2 \neq a^*, a_2 \neq a_1)$$
$$R_h(f, \cdot) := 0.$$

For $h = H$, $r_H(s, a) := \langle \phi(s, a), v(a^*) \rangle$ for every state-action pair.

**Transitions.** The initial state distribution $\mu$ is set as a uniform distribution over $\{\bar{1}, \cdots, \bar{m}\}$. The transition probabilities are set as follows.

$$\Pr[f|\overline{a_1}, a^*] = 1,$$
$$\Pr[\cdot|\overline{a_1}, a_2] = \begin{cases} \overline{a_2} : \left\langle v(a_1), v(a_2) \right\rangle + 2\gamma \\ f : 1 - \left\langle v(a_1), v(a_2) \right\rangle - 2\gamma \end{cases}, \qquad (a_2 \neq a^*, a_2 \neq a_1)$$
$$\Pr[f|f, \cdot] = 1.$$

After taking action $a_2$, the next state is either $\overline{a_2}$ or $f$. Thus this MDP looks roughly like a "leaking complete graph": starting from state $\bar{a}$, it is possible to visit any other state (except for $\overline{a^*}$); however, there is always at least $1 - 3\gamma$ probability of going to the terminal state $f$. The transition probabilities are indeed valid, because

$$0 < \gamma \leq \left\langle v(a_1), v(a_2) \right\rangle + 2\gamma \leq 3\gamma < 1.$$

We now verify that realizability, i.e. Assumption 1, is satisfied. In particular, we claim the following.

---

[3]Note that for simplicity we assume different state could have different set of available actions. In the Supplementary Material we provide another construction where all states have the same set of available actions.

**Lemma 2.** *In the MDP $\mathcal{M}_{a^*}$, $\forall h \in [H]$, for any state-action pair $(s, a)$, $Q_h^*(s, a) = \langle \phi(s, a), v(a^*) \rangle$.*

The lemma can be proved via induction, with the hypothesis being for all $a_1 \in [m]$, $a_2 \neq a_1$,

$$Q_h^*(\overline{a_1}, a_2) = \left( \left\langle v(a_1), v(a_2) \right\rangle + 2\gamma \right) \cdot \left\langle v(a_2), v(a^*) \right\rangle, \tag{1}$$

and that for all $a_1 \neq a^*$,

$$V_h^*(\overline{a_1}) = Q_h^*(\overline{a_1}, a^*) = \left\langle v(a_1), v(a^*) \right\rangle + 2\gamma. \tag{2}$$

From Eq. (1) and (2), it is easy to see that at state $\overline{a_1} \neq \overline{a^*}$, for $a_2 \neq a^*$, the suboptimality gap is

$$\Delta_h(\overline{a_1}, a_2) := V_h^*(\overline{a_1}) - Q_h^*(\overline{a_1}, a_2) > \gamma - 3\gamma^2 \geq \frac{1}{4}\gamma.$$

Thus in this MDP, Assumption 2 is satisfied with $\Delta_{\min} \geq \frac{1}{4}\gamma = \Omega(1)$. [4]

## 4.2 The information-theoretic argument

Now we are ready to state and prove our main technical lemma.

**Lemma 3.** *For any algorithm, there exists $a^* \in [m]$ such that in order to output $\pi$ with*

$$\mathbb{E}_{s_1 \sim \mu} V^\pi(s_1) \geq \mathbb{E}_{s_1 \sim \mu} V^*(s_1) - 0.05$$

*with probability at least $0.1$ for $\mathcal{M}_{a^*}$, the number of samples required is $2^{\Omega(\min\{d, H\})}$.*

We provide a proof sketch for the lower bound below. The full proof can be found in the Supplementary Material. Our main result, Theorem 1, is a direct consequence of Lemma 3.

**Proof sketch.** Observe that the feature map of $\mathcal{M}_{a^*}$ does not depend on $a^*$, and that for $h < H$ and $a_2 \neq a^*$, the reward $R_h(\overline{a_1}, a_2)$ also contains no information about $a^*$. The transition probabilities are also independent of $a^*$, unless the action $a^*$ is taken. Moreover, the reward at state $f$ is always $0$. Thus, to receive information about $a^*$, the agent either needs to take the action $a^*$, or be at a non-terminal state at the final time step ($h = H$).

However, note that the probability of remaining at a non-terminal state at the next layer is at most

$$\sup_{a_1 \neq a_2} \langle v(a_1), v(a_2) \rangle + 2\gamma \leq 3\gamma \leq \frac{3}{4}.$$

Thus for any algorithm, $\Pr[s_H \neq f] \leq \left(\frac{3}{4}\right)^H$, which is exponentially small.

In other words, any algorithm that does not know $a^*$ either needs to "be lucky" so that $s_H = f$, or needs to take $a^*$ "by accident". Since the number of actions is $m = 2^{\Theta(d)}$, either event cannot happen with constant probability unless the number of episodes is exponential in $\min\{d, H\}$.

In order to make this claim rigorous, we can construct a reference MDP $\mathcal{M}_0$ as follows. The state space, action space, and features of $\mathcal{M}_0$ are the same as those of $\mathcal{M}_a$. The transitions are defined as follows:

$$\Pr[\cdot | \overline{a_1}, a_2] = \begin{cases} \overline{a_2} : \left\langle v(a_1), v(a_2) \right\rangle + 2\gamma \\ f : 1 - \left\langle v(a_1), v(a_2) \right\rangle - 2\gamma \end{cases}, \qquad (\forall a_1, a_2 \text{ s.t. } a_1 \neq a_2)$$

$$\Pr[f | f, \cdot] = 1.$$

The rewards are defined as follows:

$$R_h(\overline{a_1}, a_2) := -2\gamma \left[ \left\langle v(a_1), v(a_2) \right\rangle + 2\gamma \right], \qquad (\forall a_1, a_2 \text{ s.t. } a_1 \neq a_2)$$

$$R_h(f, \cdot) := 0.$$

---

[4]Here we ignored the terminal state $f$ and the essentially unreachable state $\overline{a^*}$ for simplicity. This issue will be handled in the Supplementary Material rigorously.

Note that $\mathcal{M}_0$ is identical to $\mathcal{M}_{a^*}$, except when $a^*$ is taken, or when an trajectory ends at a non-terminal state. Since the latter event happens with an exponentially small probability, we can show that for any algorithm, the probability of taking $a^*$ in $\mathcal{M}_{a^*}$ is close to the probability of taking $a^*$ in $\mathcal{M}_0$. Since $\mathcal{M}_0$ is independent of $a^*$, unless an exponential number of samples are used, for any algorithm there exists $a^* \in [m]$ such that the probability of taking $a^*$ in $\mathcal{M}_0$ is $o(1)$. It then follows that the probability of taking $a^*$ in $\mathcal{M}_{a^*}$ is $o(1)$. Since $a^*$ is the optimal action for every state, such an algorithm cannot output a near-optimal policy for $\mathcal{M}_{a^*}$.

## 5   Upper Bounds under Further Assumptions

Theorem 1 suggests that Assumption 1 and Assumption 2 are not sufficient for sample-efficient RL when the number of actions could be exponential, and that additional assumptions are needed to achieve polynomial sample complexity. One style of assumption is via assuming a global representation property on the features, such as completeness [Zanette et al., 2020].

In this section, we consider two assumptions on additional structures on the transitions of the MDP rather than the feature representation that enable good rates for linear regression with sparse bias. The first condition is a variant of the low variance condition in Du et al. [2019b].

**Assumption 3** (Low variance condition)**.** There exists a constant $1 \leq C_{\text{var}} < \infty$ such that for any $h \in [H]$ and any policy $\pi$,

$$\mathbb{E}_{s \sim \mathcal{D}_h^\pi} \left[ |V^\pi(s) - V^*(s)|^2 \right] \leq C_{\text{var}} \cdot \left( \mathbb{E}_{s \sim \mathcal{D}_h^\pi} \left[ |V^\pi(s) - V^*(s)| \right] \right)^2.$$

The second assumption is that the feature distribution is hypercontractive.

**Assumption 4.** There exists a constant $1 \leq C_{\text{hyper}} < \infty$ such that for any $h \in [H]$ and any policy $\pi$, the distribution of $\phi(s, a)$ with $(s, a) \sim \mathcal{D}_h^\pi$ is $(C_{\text{hyper}}, 4)$-hypercontractive. In other words, $\forall \pi$, $\forall h \in [H], \forall v \in \mathbb{R}^d$,

$$\mathbb{E}_{(s,a) \sim \mathcal{D}_h^\pi} \left[ (\phi(s, a)^\top v)^4 \right] \leq C_{\text{hyper}} \cdot \left( \mathbb{E}_{(s,a) \sim \mathcal{D}_h^\pi} [(\phi(s, a)^\top v)^2] \right)^2.$$

Intuitively, hypercontractivity characterizes the anti-concentration of a distribution. A broad class of distributions are hypercontractive with $C_{\text{hyper}} = O(1)$, including Gaussian distributions (of arbitrary covariance matrices), uniform distributions over the hypercube and sphere, and strongly log-concave distributions [Kothari and Steurer, 2017]. Hypercontractivity has been previously used for outlier-robust linear regression [Klivans et al., 2018, Bakshi and Prasad, 2020] and moment-estimation [Kothari and Steurer, 2017].

We show that under Assumptions 1, 2, 3 or 1, 2, 4, a modified version of the Difference Maximization Q-learning (DMQ) algorithm [Du et al., 2019b] is able to learn a near-optimal policy using polynomial number of trajectories with no dependency on the number of actions.

### 5.1   Optimal experiment design

Given a set of $d$-dimensional vectors, $G$-optimal experiment design aims at finding a distribution $\rho$ over the vectors such that when sampling from this distribution, the maximum prediction variance over the set via linear regression is minimized. The following lemma on G-optimal design is a direct corollary of the Kiefer-Wolfowitz theorem [Kiefer and Wolfowitz, 1960].

**Lemma 4** (Existence of G-optimal design)**.** *For any set $X \subseteq \mathbb{R}^d$, there exists a distribution $\rho_X$ supported on $X$, known as the G-optimal design, such that*

$$\max_{x \in X} x^\top \left( \mathbb{E}_{z \sim \rho_X} z z^\top \right)^{-1} x \leq d.$$

Efficient algorithms for finding such a distribution can be found in Todd [2016].

In the context of reinforcement learning, the set $X$ corresponds to the set of all features, which is inaccessible. Instead, one can only observe one state $s$ at a time, and choose $a \in \mathcal{A}$ based on the features $\{\phi(s, a)\}_{a \in \mathcal{A}}$. Such a problem is closer to the distributional optimal design problem described by Ruan et al. [2020]. For our purpose, the following simple approach suffices: given a state

$s$, perform exploration by sampling from the G-optimal design on $\{\phi(s,a)\}_{a\in\mathcal{A}}$. The performance of this exploration strategy is guaranteed by the following lemma, which will be used in the analysis of Algorithm 1.

**Lemma 5** (Lemma 4 in Ruan et al. [2020]). *For any state $s$, denote the G-optimal design with its features by $\rho_s(\cdot) \in \Delta_A$, and the corresponding covariance matrix by $\Sigma_s := \sum_a \rho_s(a)\phi(s,a)\phi(s,a)^\top$. Given a distribution $\nu$ over states. Denote the average covariance matrix by $\Sigma := \mathbb{E}_{s\sim\nu}\Sigma_s$. Then*

$$\mathbb{E}_{s\sim\nu}\left[\max_{a\in\mathcal{A}} \phi(s,a)^\top \Sigma^{-1}\phi(s,a)\right] \leq d^2.$$

Note that the performance of this strategy is only worse by a factor of $d$ (compared to the case where one can query all features), and has no dependency on the number of actions.

## 5.2 The modified DMQ algorithm

**Overview.** During the execution of the Difference Maximization Q-learning (DMQ) algorithm, for each level $h \in [H]$, we maintain three variables: the estimated linear coefficients $\theta_h \in \mathbb{R}^d$, a set of exploratory policies $\Pi_h$, and the empirical feature covariance matrix $\Sigma_h$ associated with $\Pi_h$. We initialize $\theta_h = \mathbf{0} \in \mathbb{R}^d$, $\Sigma_h := \lambda_r I_{d\times d}$ and $\Pi_h$ to as a single purely random exploration policy, i.e., $\Pi_h = \{\pi\}$ where $\pi$ chooses an action uniformly at random for all states.[5]

Each time we execute Algorithm 1, the goal is to update the estimated linear coefficients $\theta_h \in \mathbb{R}^d$, so that for all $\pi \in \Pi_h$, $\theta_h$ is a good estimation to $\theta_h^*$ with respect to the distribution induced by $\pi$. We run ridge regression on the data distribution induced by policies in $\Pi_h$, and the regression targets are collected by invoking the greedy policy induced by $\{\theta_{h'}\}_{h'>h}$.

However, there are two apparent issues with such an approach. First, for levels $h' > h$, $\theta_{h'}$ is guaranteed to achieve low estimation error only with respect to the distributions induced by policies $\Pi_{h'}$. It is possible that for some $\pi \in \Pi_h$, the estimation error of $\theta_{h'}$ is high for the distribution induced by $\pi$ (followed by the greedy policy). To resolve this issue, the main idea in Du et al. [2019b] is to explicitly check whether $\theta_{h'}$ also predicts well on the new distribution (see Line 5 in Algorithm 1). If not, we add the new policy into $\Pi_{h'}$ and invoke Algorithm 1 recursively. The analysis in Du et al. [2019b] upper bounds the total number of recursive calls by a potential function argument, which also gives an upper bound on the sample complexity of the algorithm.

Second, the exploratory policies $\Pi_h$ only induce a distribution over states at level $h$, and the algorithm still needs to decide an exploration strategy to choose actions at level $h$. To this end, the algorithm in Du et al. [2019b] explores all actions uniformly at random, and therefore the sample complexity has at least linear dependency on the number of actions. We note that similar issues also appear in the linear contextual bandit literature [Lattimore and Szepesvári, 2020, Ruan et al., 2020], and indeed our solution here is to explore by sampling from the G-optimal design over the features at a single state. As shown by Lemma 5, for all possible roll-in distributions, such an exploration strategy achieves a nice coverage over the feature space, and is therefore sufficient for eliminating the dependency on the size of the action space.

**The algorithm.** The formal description of the algorithm is given in Algorithm 1. The algorithm should be run by calling LearnLevel on input $h = 0$.

Here, for a policy $\pi_h \in \Pi_h$, the associated exploratory policy $\tilde{\pi}_h$ is defined as

$$\tilde{\pi}_h(s_{h'}) = \begin{cases} \pi(s_{h'}) & \text{(if } h' < h) \\ \text{Sample from } \rho_{s_h}(\cdot) & \text{(if } h' = h) \\ \arg\max_a \phi_{h'}(s_{h'},a)^\top \theta_{h'} & \text{(if } h' > h) \end{cases} \tag{3}$$

Here $\rho_s(\cdot)$ is the G-optimal design on the set of vectors $\{\phi(s,\cdot)\}_{a\in\mathcal{A}}$, as defined by Lemma 4. Note that when $h = 0$, $\tilde{\pi}_h$ is always the greedy policy on $\{\theta_h\}_{h\in[H]}$. The choice of the algorithmic parameters $(\beta, \lambda_r, \lambda_{\text{ridge}})$ can be found in the proof of Theorem 2.

---

[5]We also define a special $\Pi_0$ in the same manner. Choice of $\lambda_r$ and other parameters can be found in the Supplementary Material.

**Algorithm 1:** LearnLevel($h$)

**Input:** A level $h \in \{0, \cdots, H\}$

1  **for** $\pi_h \in \Pi_h$ **do**
2     **for** $h' = H, H-1, \cdots, h+1$ **do**
3         Collect $N$ samples $\{(s^j_{h'}, a^j_{h'})\}_{j \in [N]}$ with $s^j_{h'} \sim \mathcal{D}^{\tilde{\pi}_h}_{h'}$ and $a^j_{h'} \sim \rho_{s^j_{h'}}$, ($\tilde{\pi}_h$ defined in (3))
4         $\hat{\Sigma}_{h'} \leftarrow \frac{1}{N} \sum_{j=1}^N \phi(s^j_{h'}, a^j_{h'}) \phi(s^j_{h'}, a^j_{h'})^\top$
5         **if** $\|\Sigma^{-\frac{1}{2}}_{h'} \hat{\Sigma}_{h'} \Sigma^{-\frac{1}{2}}_{h'}\|_2 > \beta |\Pi_{h'}|$ **then**
6            $\Pi_{h'} \leftarrow \Pi_{h'} \cup \{\tilde{\pi}_h\}$
7            LearnLevel($h'$)
8            LearnLevel($h$)

9  **if** $h = 0$ **then**
10    Output greedy policy with respect to $\{\theta_h\}_{h \in [H]}$ and exit
11  $\Sigma_h \leftarrow \frac{\lambda_r}{|\Pi_h|} I, \quad w_h \leftarrow \mathbf{0} \in \mathbb{R}^d$
12  **for** $i = 1, \cdots, N|\Pi_h|$ **do**
13    Sample $\pi$ from uniform distribution over $\Pi_h$
14    Execute $\tilde{\pi}_h$ (see (3)) to collect $(s^i_h, a^i_h, y_i)$, where $y_i := \sum_{h' \geq h} r^i_{h'}$ is the on-the-go reward
15    $\Sigma_h \leftarrow \Sigma_h + \frac{1}{N|\Pi_h|} \phi(s^i_h, a^i_h) \phi(s^i_h, a^i_h)^\top$
16    $w_h \leftarrow w_h + \frac{1}{N|\Pi_h|} \phi(s^i_h, a^i_h) y_i$
17  $\theta_h \leftarrow \left( (\lambda_{\text{ridge}} - \frac{\lambda_r}{|\Pi_h|}) I + \Sigma_h \right)^{-1} w_h$

## 5.3 Analysis

We show the following theorem regarding the modified algorithm.

**Theorem 2.** *Assume that Assumption 1, 2 and one of Assumption 3 and 4 hold. Also assume that*

$$\epsilon \leq \text{poly}(\Delta_{\min}, 1/C_{\text{var}}, 1/d, 1/H) \qquad \text{(Under Assumption 3)}$$
$$\text{or} \quad \epsilon \leq \text{poly}(\Delta_{\min}, 1/C_{\text{hyper}}, 1/d, 1/H). \qquad \text{(Under Assumption 4)}$$

*Let $\mu$ be the initial state distribution. Then with probability $1 - \epsilon$, running Algorithm 1 on input $0$ returns a policy $\pi$ which satisfies $\mathbb{E}_{s_1 \sim \mu} V^\pi(s_1) \geq \mathbb{E}_{s_1 \sim \mu} V^*(s_1) - \epsilon$ using $\text{poly}(1/\epsilon)$ trajectories.*

Note that here both the algorithm and the theorem have no dependence on the number of actions $A$. The proof of the theorem under Assumption 3 is largely based on the analysis in Du et al. [2019b]. The largest difference is that we used Lemma 5 instead of the original union bound argument when controlling $\Pr \left[ \sup_a |\theta_h^\top \phi(s, a) Q_h^*(s, a)| > \frac{\gamma}{2} \right]$. The proof under Assumption 4 relies on a novel analysis of least squares regression under hypercontractivity. The full proof can be found in the Supplementary Material.

## 6 Discussion

**Exponential separation between the generative model and the online setting.** When a generative model (also known as simulator) is available, Assumption 1 and Assumption 2 are sufficient for designing an algorithm with $\text{poly}(1/\epsilon, 1/\Delta_{\min}, d, H)$ sample complexity [Du et al., 2019a, Theorem C.1]. As shown by Theorem 1, under the standard online RL setting (i.e. without access to a generative model), the sample complexity is lower bounded by $2^{\Omega(\min\{d, H\})}$ when $\Delta_{\min} = \Theta(1)$, under the same set of assumptions. This implies that the generative model is *exponentially* more powerful than the standard online RL setting.

Although the generative model is conceptually much stronger than the online RL model, previously little is known on the extent to which the former is more powerful. In tabular RL, for instance, the known sample complexity bounds with or without access to generative models are nearly the same [Zhang et al., 2020, Agarwal et al., 2020]. To the best of our knowledge, the only existing example of such separation is shown by Wang et al. [2020a] under the following set of conditions: (i)

deterministic system; (ii) realizability (Assumption 1); (iii) no reward feedback (a.k.a. reward-free exploration). In comparison, our separation result holds under less restrictions (allows stochasticity) and for the usual RL environment (instead of reward-free exploration), and is thus far more natural.

**Connecting Theorem 1 and Theorem 2.** Our hardness result in Theorem 1 shows that under Assumption 1 and Assumption 2, any algorithm requires exponential number of samples to find a near-optimal policy, and therefore, sample-efficient RL is impossible without further assumptions (e.g., Assumption 3 or 4 assumed in Theorem 2). Indeed, Theorem 1 and Theorem 2 imply that the coefficients $C_{\text{var}}$ and $C_{\text{hyper}}$ in Assumption 3 and 4 are at least exponential for the hard MDP family used in Theorem 1, which can also be verified easily.

**Open problems.** The first open problem is whether a sample complexity lower bound under Assumption 1 can be shown with polynomial number of actions. This will further rule out $\text{poly}(A, d, H)$-style upper bounds, which are still possible with the current results. Another open problem is whether Assumption 3 or 4 can be replaced by or understood as more natural characterizations of the complexity of the MDP.

## Acknowledgments and Disclosure of Funding

The authors would like to thank Kefan Dong and Dean Foster for helpful discussions. Sham M. Kakade acknowledges funding from the ONR award N00014-18-1-2247 and from the National Science Foundation under award #CCF-1703574. Ruosong Wang was supported in part by the NSF IIS1763562, US Army W911NF1920104, and ONR Grant N000141812861.

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
