# Supplementary Material for: An Exponential Lower Bound for Linearly-Realizable MDPs with Constant Suboptimality Gap

## 1 Proof of Lemma 2

*Proof.* We first verify the statement for the terminal state $f$. Observe that at the terminal state $f$, regardless of the action taken, the next state is always $f$ and the reward is always $0$. Hence $Q_h^*(f, \cdot) = V_h^*(f) = 0$ for all $h \in [H]$. Thus $Q_h^*(f, \cdot) = \langle \phi(f, \cdot), v(a^*) \rangle = 0$.

We now verify realizability for other states via induction on $h = H, H-1, \cdots, 1$. The induction hypothesis is $\forall a_1 \in [m]$, $a_2 \neq a_1$,

$$Q_h^*(\overline{a_1}, a_2) = \left( \left\langle v(a_1), v(a_2) \right\rangle + 2\gamma \right) \cdot \left\langle v(a_2), v(a^*) \right\rangle, \tag{1}$$

and that $\forall a_1 \neq a^*$,

$$V_h^*(\overline{a_1}) = Q_h^*(\overline{a_1}, a^*) = \left\langle v(a_1), v(a^*) \right\rangle + 2\gamma. \tag{2}$$

When $h = H$, (1) holds by the definition of rewards. Next, note that $\forall h$, (2) follows from (1). This is because for $a_2 \neq a^*, a_1$,

$$Q_h^*(\overline{a_1}, a_2) = \left( \left\langle v(a_1), v(a_2) \right\rangle + 2\gamma \right) \cdot \left\langle v(a_2), v(a^*) \right\rangle \leq 3\gamma^2,$$

while

$$Q_h^*(\overline{a_1}, a^*) = \left\langle v(a_1), v(a^*) \right\rangle + 2\gamma \geq \gamma > 3\gamma^2.$$

In other words, (1) implies that $a^*$ is always the optimal action. Thus, it remains to show that (1) holds for $h$ assuming (2) holds for $h+1$. By Bellman's optimality equation,

$$
\begin{aligned}
Q_h^*(\overline{a_1}, a_2) &= R_h(\overline{a_1}, a_2) + \mathbb{E}_{s_{h+1}} \left[ V_{h+1}^*(s_{h+1}) \big| \overline{a_1}, a_2 \right] \\
&= -2\gamma \left[ \left\langle v(a_1), v(a_2) \right\rangle + 2\gamma \right] + \Pr[s_{h+1} = \overline{a_2}] \cdot V_{h+1}^*(a_2) + \Pr[s_{h+1} = f] \cdot V_{h+1}^*(f) \\
&= -2\gamma \left[ \left\langle v(a_1), v(a_2) \right\rangle + 2\gamma \right] + \left[ \left\langle v(a_1), v(a_2) \right\rangle + 2\gamma \right] \cdot \left( \left\langle v(a_1), v(a^*) \right\rangle + 2\gamma \right) \\
&= \left( \left\langle v(a_1), v(a_2) \right\rangle + 2\gamma \right) \cdot \left\langle v(a_1), v(a^*) \right\rangle.
\end{aligned}
$$

This is exactly (1) for $h$. Hence both (1) and (2) hold for all $h \in [H]$. $\square$

## 2 Proof of Lemma 5

*Proof.* We state a proof of this lemma for completeness. By Lemma 4, $\forall s$,

$$\max_{a \in \mathcal{A}} \phi(s, a)^\top \Sigma_s^{-1} \phi(s, a) \leq d.$$

It follows that $\forall a \in \mathcal{A}$,

$$\phi(s, a)\phi(s, a)^\top \preccurlyeq d\Sigma_s.$$

Therefore,

$$\mathbb{E}_{s\sim\nu}\left[\max_{a\in\mathcal{A}}\phi(s,a)^\top \Sigma^{-1}\phi(s,a)\right] = \mathbb{E}_{s\sim\nu}\max_{a\in\mathcal{A}}\operatorname{Tr}\left(\phi(s,a)\phi(s,a)^\top\Sigma^{-1}\right)$$
$$\leq \mathbb{E}_{s\sim\nu}\operatorname{Tr}\left(d\Sigma_s\Sigma^{-1}\right) = d^2.$$

$\square$

## 3 Addressing Footnote 3

Let us redefine $\mathcal{M}_{a^*}$ as follows. The state space is again $\{\bar{1},\cdots,\bar{m},f\}$. The action space is $[m]$ for every state. We will also use the same set of $m$ $d$-dimensional vectors $\{v_1,\cdots,v_m\}$. In this construction, we will reset $\gamma := \frac{1}{6}$.

**Features.** The feature map now maps state-action pairs to $d+1$ dimensional vectors, and is defined as follows.

$$\phi(\overline{a_1},a_2) := \left(0, \left(\left\langle v(a_1),v(a_2)\right\rangle + 2\gamma\right)\cdot v(a_2)\right), \qquad (\forall a_1,a_2\in[m], a_1\neq a_2)$$

$$\phi(\overline{a_1},a_1) := \left(\frac{3}{4}\gamma, 0\right), \qquad (\forall a_1\in[m])$$

$$\phi(f,1) = (0,\mathbf{0}),$$
$$\phi(f,a) := (-1,\mathbf{0}). \qquad (\forall a\neq 1)$$

Note that the feature map is again independent of $a^*$. Define $\theta^* := (1,v(a^*))$.

**Rewards.** For $1\leq h < H$, the rewards are defined as

$$R_h(\overline{a_1},a^*) := \left\langle v(a_1),v(a^*)\right\rangle + 2\gamma, \qquad (a_1\neq a^*)$$

$$R_h(\overline{a_1},a_2) := -2\gamma\left[\left\langle v(a_1),v(a_2)\right\rangle + 2\gamma\right], \qquad (a_2\neq a^*, a_2\neq a_1)$$

$$R_h(\overline{a_1},a_1) := \frac{3}{4}\gamma, \qquad (\forall a_1)$$

$$R_h(f,1) := 0,$$
$$R_h(f,a) := -1. \qquad (a\neq 1)$$

For $h=H$, $r_H(s,a) := \langle\phi(s,a),v(a^*)\rangle$ for every state-action pair.

**Transitions.** The initial state distribution is set as a uniform distribution over $\{\bar{1},\cdots,\bar{m}\}$. The transition probabilities are set as follows.

$$\Pr[f|\overline{a_1},a^*] = 1,$$
$$\Pr[f|\overline{a_1},a_1] = 1,$$
$$\Pr[\cdot|\overline{a_1},a_2] = \begin{cases} \overline{a_2} : \left\langle v(a_1),v(a_2)\right\rangle + 2\gamma \\ f : 1 - \left\langle v(a_1),v(a_2)\right\rangle - 2\gamma \end{cases}, \qquad (a_2\neq a^*, a_2\neq a_1)$$
$$\Pr[f|f,\cdot] = 1.$$

We now check realizability in the new MDP. Note that now we want to show $Q_h^*(s,a) = \phi(s,a)^\top\theta^*$, where $\theta^* = (1,v(a^*))$. We claim that $\forall h\in[H]$,

$$V_h^*(\overline{a_1}) = \langle v(a_1),v(a^*)\rangle + 2\gamma, \qquad (a_1\neq a^*)$$
$$Q_h^*(\overline{a_1},a_2) = (\langle v(a_1),v(a_2)\rangle + 2\gamma)\cdot\langle v(a_2),v(a^*)\rangle, \qquad (a_2\neq a_1)$$
$$Q_h^*(\overline{a_1},a_1) = \frac{3}{4}\gamma. \qquad (\forall a_1)$$

To see this, first notice that the expression of $Q_h^*$ implies that the optimal action is $a^*$ for any non-terminal state. Suppose $a_1 \neq a^*$, then for $a_2 \neq a_1, a^*$, $Q_h^*(\overline{a_1}, a_2) \leq 3\gamma^2 < \gamma \leq Q_h^*(\overline{a_1}, a^*)$. Moreover,

$$Q_h^*(\overline{a_1}, a_1) = \frac{3}{4}\gamma < \gamma \leq Q_h^*(\overline{a_1}, a^*).$$

Thus, $a^*$ is indeed the optimal action for $\overline{a_1}$ if $a_1 \neq a^*$.

For $\overline{a^*}$, $a_1 \neq a^*$, $Q_h^*(\overline{a^*}, a_1) \leq 3\gamma^2 < \frac{3}{4}\gamma = Q_h^*(\overline{a^*}, a^*)$. Therefore, $a^*$ is the optimal action for all states (besides $f$).

As for $f$, it is easy to see that $Q_h^*(f, 1) = 0$, and that $\forall a \neq 1$, $Q_h^*(f, a) = -1$.

What remains is show the statements for all $h$ via induction. Suppose that

$$Q_{h+1}^*(\overline{a_1}, a_2) = (\langle v(a_1), v(a_2) \rangle + 2\gamma) \cdot \langle v(a_2), v(a^*) \rangle. \qquad (a_2 \neq a_1)$$

Then indeed $V_{h+1}^*(\overline{a_1}) = Q_{h+1}^*(\overline{a_1}, a^*) = \langle v(a_1), v(a^*) \rangle + 2\gamma$. It follows that $\forall a_2 \neq a^*$

$$\begin{aligned}
Q_h^*(\overline{a_1}, a_2) &= R_h(\overline{a_1}, a_2) + \mathbb{E}_{s_{h+1}} \left[ V_{h+1}^*(s_{h+1}) \middle| \overline{a_1}, a_2 \right] \\
&= -2\gamma \left[ \left\langle v(a_1), v(a_2) \right\rangle + 2\gamma \right] + \Pr[s_{h+1} = \overline{a_2}] \cdot V_{h+1}^*(a_2) + \Pr[s_{h+1} = f] \cdot V_{h+1}^*(f) \\
&= -2\gamma \left[ \left\langle v(a_1), v(a_2) \right\rangle + 2\gamma \right] + \left[ \left\langle v(a_1), v(a_2) \right\rangle + 2\gamma \right] \cdot \left( \left\langle v(a_1), v(a^*) \right\rangle + 2\gamma \right) \\
&= \left( \left\langle v(a_1), v(a_2) \right\rangle + 2\gamma \right) \cdot \left\langle v(a_1), v(a^*) \right\rangle.
\end{aligned}$$

**Suboptimality Gap.** In $\mathcal{M}_{a^*}$, $\forall a_1 \neq a^*$, $\forall a_2 \neq a^*$, $Q_h^*(\overline{a_1}, a_2) \leq \max\{3\gamma^2, \frac{3}{4}\gamma\}$. Thus

$$\Delta_h(\overline{a_1}, a_2) \geq \gamma - \max\{3\gamma^2, \frac{3}{4}\gamma\} = \frac{1}{24}.$$

For $\overline{a^*}$, $V_h^*(\overline{a^*}) = 1 - \gamma$, while for $a_1 \neq a^*$,

$$Q_h^*(\overline{a^*}, a_1) = (\langle v(a^*), v(a_1) + 2\gamma) \cdot \langle v(a^*), v(a_1) \rangle \leq 3\gamma^2.$$

Thus $\Delta_h^*(\overline{a^*}, a_1) \geq \frac{3}{4}\gamma - 3\gamma^2 = \frac{1}{24}$. As for the terminal state $f$, the suboptimality gap is obviously 1. Therefore $\Delta_{\min} \geq \frac{1}{24}$ in this new MDP.

**Information theoretic arguments.** The modifications here do not affect the proof of Theorem 1. Suppose action $a_2$ is taken at state $\overline{a_1}$. If $a_1 \neq a_2$, then the behavior (transitions and rewards) would be identical to the original MDP. If $a_1 = a_2 \neq a^*$, neither the transition and the rewards depend on $a^*$. Hence, we can still construct a reference MDP as in the proof of Theorem 1, such that information on $a^*$ can only be gained by: (1) either taking $a^*$; (2) or reaching $s_H \neq f$.

## 4 Proof of Theorem 1

**Theorem 1.** *Consider an arbitrary online RL algorithm that takes the feature mapping $\phi : \mathcal{S} \times \mathcal{A} \to \mathbb{R}^d$ as input. In the online RL setting, there exists an MDP with a feature mapping $\phi$ satisfying Assumption 1 and Assumption 2 with $\Delta_{\min} = \Omega(1)$, such that the algorithm requires $\min\{2^{\Omega(d)}, 2^{\Omega(H)}\}$ samples to find a policy $\pi$ with*

$$\mathbb{E}_{s_1 \sim \mu} V^\pi(s_1) \geq \mathbb{E}_{s_1 \sim \mu} V^*(s_1) - 0.05$$

*with probability $0.1$.*

*Proof.* We consider $K$ episodes of interaction between the algorithm and the MDP $\mathcal{M}_a$. Since each trajectory is a sequence of $H$ states, we define the total number of samples as $KH$. Denote the state, the action and the reward at episode $k$ and timestep $h$ by $s_h^k$, $a_h^k$ and $r_h^k$ respectively.

Consider the following reference MDP denoted by $\mathcal{M}_0$. The state space, action space, and features of this MDP are the same as those of the MDP family. The transitions are defined as follows:

$$\Pr[\cdot | \overline{a_1}, a_2] = \begin{cases} \overline{a_2} : \left\langle v(a_1), v(a_2) \right\rangle + 2\gamma \\ f : 1 - \left\langle v(a_1), v(a_2) \right\rangle - 2\gamma \end{cases}, \qquad (\forall a_1, a_2 \text{ s.t. } a_1 \neq a_2)$$

$$\Pr[f | f, \cdot] = 1.$$

The rewards are defined as follows:

$$R_h(\overline{a_1}, a_2) := -2\gamma \left[ \Big\langle v(a_1), v(a_2) \Big\rangle + 2\gamma \right], \qquad (\forall a_1, a_2 \text{ s.t. } a_1 \neq a_2)$$

$$R_h(f, \cdot) := 0.$$

Intuitively, this MDP is very similar to the MDP family, except that the optimal action $a^*$ is removed. More specifically, $\mathcal{M}_0$ is identical to $\mathcal{M}_a$ except when the action $a$ is taken at a non-terminal state, or when an episode ends at a non-terminal state.

More specifically, we claim that for $t < H$, $\forall s_t, a_t$ such that $a_t \neq a$,

$$\Pr_{\mathcal{M}_a}[s_{t+1}|s_t, a_t] = \Pr_{\mathcal{M}_0}[s_{t+1}|s_t, a_t],$$

and that for $t < H$, $\forall s_t, a_t$ such that $a_t \neq a$,

$$r_t^{\mathcal{M}_a}(s_t, a_t) = r_t^{\mathcal{M}_0}(s_t, a_t).$$

Also, $r_H^{\mathcal{M}_a}(s_t, a_t) = r_H^{\mathcal{M}_0}(s_t, a_t)$ if $s_t = f$. It follows that

$$\Pr_{\mathcal{M}_a} \left[ s_1^1, a_1^1, r_1^1, \cdots s_h^k, a_h^k, r_h^k \,\Big|\, a \notin A_h^k, \forall k' \leq k, s_H^{k'} = f \right]$$

$$= \Pr_{\mathcal{M}_0} \left[ s_1^1, a_1^1, r_1^1, \cdots s_h^k, a_h^k, r_h^k \,\Big|\, a \notin A_h^k, \forall k' \leq k, s_H^{k'} = f \right].$$

Here $A_h^k$ is a shorthand for $\{a_1^1, a_2^1, \cdots, a_H^1, \cdots, a_h^k\}$, i.e. all actions taken up to timestep $h$ for episode $k$. By marginalizing the states and the actions, we get

$$\Pr_{\mathcal{M}_a} \left[ a_h^k \,\Big|\, a \notin A_h^k, \forall k' \leq k, s_H^{k'} = f \right] = \Pr_{\mathcal{M}_0} \left[ a_h^k \,\Big|\, a \notin A_h^k, \forall k' \leq k, s_H^{k'} = f \right].$$

It then follows that

$$\Pr_{\mathcal{M}_a} \left[ a_h^k = a \,\Big|\, a \notin A_h^k, \forall k' \leq k, s_H^{k'} = f \right] = \Pr_{\mathcal{M}_0} \left[ a_h^k = a \,\Big|\, a \notin A_h^k, \forall k' \leq k, s_H^{k'} = f \right].$$

Next, we prove via induction that

$$\Pr_{\mathcal{M}_a} \left[ a \in A_h^k \,\Big|\, \forall k' \leq k, s_H^{k'} = f \right] = \Pr_{\mathcal{M}_0} \left[ a \in A_h^k \,\Big|\, \forall k' \leq k, s_H^{k'} = f \right]. \qquad (3)$$

Suppose that (3) holds up to $(k, h-1)$. Then

$$\Pr_{\mathcal{M}_a} \left[ a \in A_h^k \,\Big|\, \forall k' \leq k, s_H^{k'} = f \right]$$

$$= \Pr_{\mathcal{M}_a} \left[ a \notin A_{h-1}^k \right] \Pr_{\mathcal{M}_a} \left[ a_h^k = a \,\Big|\, a \notin A_{h-1}^k, \forall k' \leq k, s_H^{k'} = f \right] + \Pr_{\mathcal{M}_a} \left[ a \in A_{h-1}^k \,\Big|\, \forall k' \leq k, s_H^{k'} = f \right]$$

$$= \Pr_{\mathcal{M}_0} \left[ a \notin A_{h-1}^k \right] \Pr_{\mathcal{M}_0} \left[ a_h^k = a \,\Big|\, a \notin A_{h-1}^k, \forall k' \leq k, s_H^{k'} = f \right] + \Pr_{\mathcal{M}_0} \left[ a \in A_{h-1}^k \,\Big|\, \forall k' \leq k, s_H^{k'} = f \right]$$

$$= \Pr_{\mathcal{M}_0} \left[ a \in A_h^k \,\Big|\, \forall k' \leq k, s_H^{k'} = f \right].$$

That is, (3) holds for $h, k$ as well. By induction, (3) holds for all $h, k$. Thus,

$$\Pr_{\mathcal{M}_a} \left[ a \in A_h^k \right] \leq \Pr_{\mathcal{M}_a} \left[ a \in A_h^k \,\Big|\, \forall k' \leq k, s_H^{k'} = f \right] + \Pr \left[ \exists k' \leq k, s_H^{k'} \neq f \right]$$

$$\leq \Pr_{\mathcal{M}_0} \left[ a \in A_h^k \,\Big|\, \forall k' \leq k, s_H^{k'} = f \right] + k \cdot \left( \frac{3}{4} \right)^H.$$

Since $|A_h^k| \leq kH$, $\sum_{a \in [m]} \Pr_{\mathcal{M}_0} \left[ a \in A_h^k \,\Big|\, \forall k' \leq k, s_H^{k'} = f \right] \leq kH$. It follows that there exists $a^* \in [m]$ such that

$$\Pr_{\mathcal{M}_0} \left[ a^* \in A_H^K \,\Big|\, \forall k' \leq K, s_H^{k'} = f \right] \leq \frac{KH}{m} = KH \cdot e^{-\Theta(d)}.$$

As a result

$$\Pr_{\mathcal{M}_{a^*}} \left[ a^* \in A_H^K \right] \leq KH \cdot e^{-\Theta(d)} + K \left( \frac{3}{4} \right)^H.$$

In other words, unless $KH = 2^{\Omega(\min\{d,H\})}$, the probability of taking the optimal action $a^*$ in the interaction with $\mathcal{M}_{a^*}$ is $o(1)$.

From the suboptimality gap condition, it follows that if $\mathbb{E}_{s_1 \sim \mu} V^\pi(s_1) \geq \mathbb{E}_{s_1 \sim \mu} V^*(s_1) - 0.05$, $\Pr\left[a_1 \neq a^* \wedge s_1 \neq \overline{a^*}\right] \cdot \Delta_{\min} \leq 0.05$. Hence

$$\Pr\left[a_1 = a^*\right] \geq 1 - \left(0.8 + \frac{1}{m}\right) = 0.2 - \frac{1}{m}.$$

Therefore, if the algorithm is able to output such a policy with probability $0.1$, it is able to take the action $a^*$ in the next episode with $\Theta(1)$ probability by executing $\pi$. However, as proved above, this is impossible unless $KH = 2^{\Omega(\min\{d,H\})}$. $\qquad\square$

## 5  Proof of Theorem 2

Recall the statements of Assumptions 3 and 4.

**Assumption 3** (Low variance condition). There exists a constant $1 \leq C_{\text{var}} < \infty$ such that for any $h \in [H]$ and any policy $\pi$,

$$\mathbb{E}_{s \sim \mathcal{D}_h^\pi}\left[|V^\pi(s) - V^*(s)|^2\right] \leq C_{\text{var}} \cdot \left(\mathbb{E}_{s \sim \mathcal{D}_h^\pi}\left[|V^\pi(s) - V^*(s)|\right]\right)^2.$$

**Assumption 4.** There exists a constant $1 \leq C_{\text{hyper}} < \infty$ such that for any $h \in [H]$ and any policy $\pi$, the distribution of $\phi(s,a)$ with $(s,a) \sim \mathcal{D}_h^\pi$ is $(C_{\text{hyper}}, 4)$-hypercontractive. In other words, $\forall \pi$, $\forall h \in [H], \forall v \in \mathbb{R}^d$,

$$\mathbb{E}_{(s,a) \sim \mathcal{D}_h^\pi}\left[(\phi(s,a)^\top v)^4\right] \leq C_{\text{hyper}} \cdot \left(\mathbb{E}_{(s,a) \sim \mathcal{D}_h^\pi}[(\phi(s,a)^\top v)^2]\right)^2.$$

**Theorem 2.** *Assume that Assumption 1, 2, and one of Assumption 3 and 4 hold. Also assume that*

$$\epsilon \leq \text{poly}(\Delta_{\min}, 1/C_{\text{var}}, 1/d, 1/H) \qquad \text{(Under Assumption 3)}$$
$$\text{or} \quad \epsilon \leq \text{poly}(\Delta_{\min}, 1/C_{\text{hyper}}, 1/d, 1/H). \qquad \text{(Under Assumption 4)}$$

*Let $\mu$ be the initial state distribution. Then with probability $1 - \epsilon$, running Algorithm 1 on input $0$ returns a policy $\pi$ which satisfies $\mathbb{E}_{s_1 \sim \mu} V^\pi(s_1) \geq \mathbb{E}_{s_1 \sim \mu} V^*(s_1) - \epsilon$ using $\text{poly}(1/\epsilon)$ trajectories.*

*Proof under Assumption 3.* Let us set $\beta = 8$, $\lambda_{\text{ridge}} = \epsilon^2$, $\lambda_r = \epsilon^6$, $B = 2d\log(\frac{d}{\lambda_r})$, $\epsilon_1 = \epsilon^2$, $\epsilon_2 = \frac{\lambda_r}{2B}$, $N = \frac{d \cdot \log(1/\epsilon_2)}{\epsilon_2^2}$. Recall that $\epsilon \leq \text{poly}(\Delta_{\min}, 1/C_{\text{var}}, 1/d, 1/H)$. First, by Lemma 8, the event $\Omega$ holds with probability $1 - \epsilon$; we will condition on this event in the following proof. By lemma 10, when the algorithm terminates, $|\Pi_h| \leq B$ for all $h \in [H]$. Note that the this implies that Algorithm 1 is called or restarted at most $H \cdot (1 + B)$ times. In each call or restart of Algorithm 1, at most $NB + N$ trajectories are sampled. Therefore, when the algorithm terminates, at most

$$H(1 + B) \cdot (NB + N) \leq \text{poly}\left(1/\epsilon\right)$$

trajectories are sampled.

It remains to show that the greedy policy with respect to $\theta_1, \cdots, \theta_H$ is indeed $\epsilon$-optimal with high probability. To that end, let us state the following claims about the algorithm.

1. Each time Line 9 is reached in Algorithm 1, $\forall \pi \in \Pi_h$, define $\tilde{\pi}_h$ as in (6), $\forall h' > h$,

$$\mathbb{E}_{s_{h'} \sim \mathcal{D}_{h'}^{\tilde{\pi}_h}}\left[\sup_{a \in \mathcal{A}} \left|\phi(s_{h'}, a)^\top(\theta_{h'} - \theta_{h'}^*)\right|^2\right] \leq \frac{\Delta_{\min}^2 \epsilon}{4H}. \qquad (4)$$

2. Each time when $\theta_h$ is updated at Line 17, $\forall \pi \in \Pi_h$, define the associated covariance matrix at step $h$ as $\Sigma_h^\pi = \mathbb{E}_{s_h \sim \mathcal{D}_h^\pi, a_h \sim \rho_{s_h}}\left[\phi(s_h, a_h)\phi(s_h, a_h)^\top\right]$. Then $\|\theta_h - \theta_h^*\|_{\Sigma_h^\pi}^2 \leq 6BC_{\text{var}}\epsilon^2$. It follows that

$$\mathbb{E}_{s_h \sim \mathcal{D}_h^\pi}\left[\sup_{a \in \mathcal{A}} \left|\phi(s_h, a)^\top(\theta_h - \theta_h^*)\right|^2\right] \leq \frac{\Delta_{\min}^2 \epsilon}{4H}. \qquad (5)$$

Note that by the first claim with $h = 0$, it follows that for the greedy policy $\hat{\pi}$ ($\tilde{\pi}_0$ is always the greedy policy) w.r.t. $\{\theta_h\}_{h \in [H]}$, $\forall h \in [H]$,

$$\mathbb{E}_{s_h \sim \mathcal{D}_h^{\hat{\pi}}} \left[ \sup_{a \in \mathcal{A}} \left| \phi(s_h, a)^\top (\theta_h - \theta_h^*) \right|^2 \right] \leq \frac{\Delta_{\min}^2 \epsilon}{4H}.$$

Consequently by Markov's inequality,

$$\Pr_{s_h \sim \mathcal{D}_h^{\hat{\pi}}} \left[ \exists a \in \mathcal{A} : \left| \phi(s_h, a)^\top (\theta_h - \theta_h^*) \right| > \frac{\Delta_{\min}}{2} \right] \leq \frac{\epsilon}{H}.$$

By Assumption 2 and the fact that $\hat{\pi}$ takes the greedy action w.r.t. $\theta_h$, this implies that

$$\Pr_{s_h \sim \mathcal{D}_h^{\hat{\pi}}} \left[ \hat{\pi}_h(s_h) \neq \pi_h^*(s_h) \right] \leq \frac{\epsilon}{H}.$$

Thus for a random trajectory induced by $\hat{\pi}$, with probability at least $1 - \epsilon$, $\hat{\pi}_h(s_h) = \pi_h^*(s_h)$ for all $h = 1, \cdots, H$, which proves the theorem.

It remains to prove the two claims.

**Proof of (5).** We first prove the second claim based on the assumption that the first claim holds when Line 9 is reached in the same execution of LearnLevel. By the first claim and the same arguments above, $\forall \pi \in \Pi_h$, construct $\tilde{\pi}_h$ as

$$\tilde{\pi}_h(s_{h'}) = \begin{cases} \pi(s_{h'}) & (\text{if } h' < h) \\ \text{Sample from } \rho_{s_h}(\cdot) & (\text{if } h' = h) \\ \arg\max_a \phi_{h'}(s_{h'}, a)^\top \theta_{h'} & (\text{if } h' > h) \end{cases}, \tag{6}$$

then $\Pr_{s_{h'} \sim \mathcal{D}_{h'}^{\tilde{\pi}_h}} \left[ \tilde{\pi}_h(s_{h'}) \neq \pi^*(s_{h'}) \right] \leq \epsilon/H$. Thus,

$$\mathbb{E}_{s_{h+1} \sim \mathcal{D}_{h+1}^{\tilde{\pi}_h}} \left[ V_{h+1}^{\tilde{\pi}_h}(s_{h+1}) \right] \geq \mathbb{E}_{s_{h+1} \sim \mathcal{D}_{h+1}^{\tilde{\pi}_h}} \left[ V_{h+1}^*(s_{h+1}) \right] - \epsilon.$$

By Assumption 3, this suggests that

$$\mathbb{E}_{s_{h+1} \sim \mathcal{D}_{h+1}^{\tilde{\pi}_h}} \left[ \left( V_{h+1}^{\tilde{\pi}_h}(s_{h+1}) - V_{h+1}^*(s_{h+1}) \right)^2 \right] \leq C_{\text{var}} \epsilon^2.$$

When $(s_h, a_h, y)$ is sampled,

$$\mathbb{E}[y | s_h, a_h] = \mathbb{E}\left[ R(s_h, a_h) + V_{h+1}^{\tilde{\pi}_h}(s_{h+1}) | s_h, a_h \right]$$
$$= Q^*(s_h, a_h) + \mathbb{E}\left[ V_{h+1}^{\tilde{\pi}_h}(s_{h+1}) - V_{h+1}^*(s_{h+1}) | s_h, a_h \right],$$

where the expectation is over trajectories induced by $\tilde{\pi}_h$. In other words, $y_i := \sum_{h' \geq h} r_h^i$ can be written as $\phi(s_h^i, a_h^i)^\top \theta_h^* + b_i + \xi_i$, where $\xi_i$ is mean-zero independent noise with $|\xi_i| \leq 2$ almost surely and $b_i := \sum_{h' > h} r_{h'}^i - V_{h+1}^*(s_{h+1}^i)$ satisfies $\mathbb{E}[b_i^2] \leq C_{\text{var}} \epsilon^2$. Note that $\theta_h$ is the ridge regression estimator for this linear model. By Lemma 7,

$$\mathbb{E}_{\pi \sim \text{Unif}(\Pi_h), s_h \sim \mathcal{D}_h^\pi, a_h \sim \rho_{s_h}} \left[ \left| \phi(s_h, a_h)^\top (\theta_h - \theta_h^*) \right|^2 \right] \leq 4(C_{\text{var}} \epsilon^2 + \epsilon_1 + \lambda_{\text{ridge}}) \leq 6 C_{\text{var}} \epsilon^2.$$

It follows that $\forall \pi \in \Pi_h$,

$$\mathbb{E}_{s_h \sim \mathcal{D}_h^\pi, a_h \sim \rho_{s_h}} \left[ \left| \phi(s_h, a_h)^\top (\theta_h - \theta_h^*) \right|^2 \right] \leq |\Pi_h| \cdot 6 C_{\text{var}} \epsilon^2 \leq 6 B C_{\text{var}} \epsilon^2.$$

Now, by Lemma 5,

$$\mathbb{E}_{s_h \sim \mathcal{D}_h^\pi} \left[ \sup_{a \in \mathcal{A}} \left| \phi(s_h, a)^\top (\theta_h - \theta_h^*) \right|^2 \right]$$
$$\leq \mathbb{E}_{s_h \sim \mathcal{D}_h^\pi} \left[ \sup_{a \in \mathcal{A}} \| \phi(s_h, a) \|_{(\Sigma_h^\pi)^{-1}}^2 \right] \cdot \| \phi_h - \phi_h^* \|_{\Sigma_h^\pi}^2$$
$$\leq d^2 \cdot 6 B C_{\text{var}} \epsilon^2 \leq \frac{\Delta_{\min}^2 \epsilon}{4H}.$$

This proves the second claim.

**Proof of (4).** Now, let us prove the first claim, assuming that the second claim holds for the last update of any $\theta_h$. By observing Algorithm 1, if Line 9 is reached, during the last execution of the first `for` loop (i.e. Lines 1 to 8), the `if` clause at Line 5 must have returned False every time (otherwise the algorithm will restart). It follows that during the last execution of Lines 1 to 8, neither $\{\theta_h\}_{h\in[H]}$ nor $\{\Pi_h\}_{h\in[H]}$ is updated.

Consider the `if` clause when checking $\pi \in \Pi_h$ for layer $h'$. Recall that

$$\Sigma_{h'}^{\tilde{\pi}_h} = \mathbb{E}_{s_{h'}\sim\mathcal{D}_{h'}^{\tilde{\pi}_h},a_{h'}\sim\rho_{s_{h'}}}\left[\phi(s_{h'},a_{h'})\phi(s_{h'},a_{h'})^\top\right].$$

Also define $\Sigma_{h'}^* := \frac{\lambda_r}{|\Pi_{h'}|}I + \mathbb{E}_{\pi\sim\text{Unif}(\Pi_{h'})}\Sigma_{h'}^\pi$. Then by Lemma 9,

$$\left\| (\Sigma_{h'}^*)^{-\frac{1}{2}}\, \Sigma_{h'}^{\tilde{\pi}_h}\, (\Sigma_{h'}^*)^{-\frac{1}{2}} \right\|_2 \le 3\beta|\Pi_{h'}|.$$

It follows that

$$\begin{aligned}
\|\theta_{h'} - \theta_{h'}^*\|_{\Sigma_{h'}^{\tilde{\pi}_h}}^2 &= (\theta_{h'} - \theta_{h'}^*)^\top \Sigma_{h'}^{\tilde{\pi}_h} (\theta_{h'} - \theta_{h'}^*) \\
&= \left((\Sigma_{h'}^*)^{\frac{1}{2}}(\theta_{h'} - \theta_{h'}^*)\right)^\top \left((\Sigma_{h'}^*)^{-\frac{1}{2}}\,\Sigma_{h'}^{\tilde{\pi}_h}\,(\Sigma_{h'}^*)^{-\frac{1}{2}}\right)\left((\Sigma_{h'}^*)^{\frac{1}{2}}(\theta_{h'} - \theta_{h'}^*)\right) \\
&\le \|\theta_{h'} - \theta_{h'}^*\|_{\Sigma_{h'}^*}^2 \cdot \| (\Sigma_{h'}^*)^{-\frac{1}{2}}\,\Sigma_{h'}^{\tilde{\pi}_h}\,(\Sigma_{h'}^*)^{-\frac{1}{2}} \|_2 \\
&\le 3\beta B \cdot \left(\lambda_r \cdot \left(\frac{2}{\lambda_{\text{ridge}}}\right)^2 + 6BC_{\text{var}}\epsilon^2\right) \\
&\le 24B^2 \cdot 10C_{\text{var}}\epsilon^2.
\end{aligned}$$

By Lemma 5,

$$\mathbb{E}_{s_{h'}\sim\mathcal{D}_h^{\tilde{\pi}_h}}\left[\sup_{a\in\mathcal{A}}\|\phi(s_{h'},a)\|_{(\Sigma_{h'}^{\tilde{\pi}_h})^{-1}}^2\right] \le d^2.$$

As a result,

$$\mathbb{E}_{s_{h'}\sim\mathcal{D}_{h'}^{\tilde{\pi}_h}}\left[\sup_{a\in\mathcal{A}}\left|\phi(s_{h'},a)^\top(\theta_{h'} - \theta_{h'}^*)\right|^2\right] \le \mathbb{E}_{s_{h'}\sim\mathcal{D}_h^{\tilde{\pi}_h}}\left[\|\theta_{h'} - \theta_{h'}^*\|_{\Sigma_{h'}^{\tilde{\pi}_h}}^2 \cdot \sup_{a\in\mathcal{A}}\|\phi(s_{h'},a)\|_{(\Sigma_{h'}^{\tilde{\pi}_h})^{-1}}^2\right]$$

$$\le 240B^2 C_{\text{var}}\epsilon^2 \cdot d^2 \le \frac{\epsilon\Delta_{\min}^2}{4H}.$$

This proves the first claim. The failure probability of the algorithm is controlled by Lemma 8. □

*Proof under Assumption 4.* The proof under Assumption 4 is quite similar, except that we will use Lemma 14 instead of Lemma 7 for the analysis of ridge regression. The different analysis of ridge regression results in a slightly different choice of algorithmic parameters.

Let us set $\beta = 8$, $\epsilon_0 = \epsilon^2$, $\lambda_{\text{ridge}} = \epsilon^3$, $\lambda_r = \epsilon^9$, $B = 2d\log(\frac{d}{\lambda_r})$, $\epsilon_1 = \epsilon^3$, $\epsilon_2 = \frac{\lambda_r}{2B}$, $N = \frac{d}{\epsilon_2^3}$. Recall that $\epsilon \le \text{poly}(\Delta_{\min}, 1/C_{\text{hyper}}, 1/d, 1/H)$. We will state similar claims about the algorithm.

1. Each time Line 9 is reached in Algorithm 1, $\forall \pi \in \Pi_h$, define $\tilde{\pi}_h$ as in (6), $\forall h' > h$,

$$\mathbb{E}_{s_{h'}\sim\mathcal{D}_{h'}^{\tilde{\pi}_h}}\left[\sup_{a\in\mathcal{A}}\left|\phi(s_{h'},a)^\top(\theta_{h'} - \theta_{h'}^*)\right|^2\right] \le \frac{\Delta_{\min}^2\epsilon_0}{4H}. \tag{7}$$

2. Each time when $\theta_h$ is updated at Line 17, $\forall \pi \in \Pi_h$, define the associated covariance matrix at step $h$ as $\Sigma_h^\pi = \mathbb{E}_{s_h\sim\mathcal{D}_h^\pi,a_h\sim\rho_{s_h}}\left[\phi(s_h,a_h)\phi(s_h,a_h)^\top\right]$. Then $\|\theta_h - \theta_h^*\|_{\Sigma_h^\pi}^2 \le \frac{\Delta_{\min}^2\epsilon_0}{120HBd^2}$. It follows that

$$\mathbb{E}_{s_h\sim\mathcal{D}_h^\pi}\left[\sup_{a\in\mathcal{A}}\left|\phi(s_h,a)^\top(\theta_h - \theta_h^*)\right|^2\right] \le \frac{\Delta_{\min}^2\epsilon_0}{4H}. \tag{8}$$

As in the proof under Assumption 3, these two claims are sufficient to guarantee that the greedy policy induced by $\{\theta_h\}_{h\in[H]}$ is $\epsilon$-optimal. We now prove the two claims in similar fashion.

**Proof of (8).** We first prove the second claim based on the assumption that the first claim holds when Line 9 is reached in the same execution of LearnLevel. By the first claim, $\forall \pi \in \Pi_h$, construct $\tilde{\pi}_h$ as in (6), then

$$\Pr_{s_{h'} \sim \mathcal{D}_{h'}^{\tilde{\pi}_h}} \left[ \tilde{\pi}_h(s_{h'}) \neq \pi^*(s_{h'}) \right] \leq \epsilon_0/H. \tag{9}$$

When $(s_h, a_h, y)$ is sampled,

$$\mathbb{E}\left[y|s_h, a_h\right] = \mathbb{E}\left[R(s_h, a_h) + V_{h+1}^{\tilde{\pi}_h}(s_{h+1})|s_h, a_h\right]$$

$$= Q^*(s_h, a_h) + \mathbb{E}\left[V_{h+1}^{\tilde{\pi}_h}(s_{h+1}) - V_{h+1}^*(s_{h+1})|s_h, a_h\right],$$

where the expectation is over trajectories induced by $\tilde{\pi}_h$. In other words, $y_i := \sum_{h' \geq h} r_h^i$ can be written as $\phi(s_h^i, a_h^i)^\top \theta_h^* + b_i + \xi_i$, where $\xi_i$ is mean-zero independent noise with $|\xi_i| \leq 2$ almost surely, and $b_i$ is defined as

$$b_i := -\sum_{h' > h} \left(V^*(s_{h'}^i) - Q^*(s_{h'}^i, a_{h'}^i)\right).$$

Here $\mathbb{E}[\xi_i] = 0$ because

$$\mathbb{E}[\xi_i] = \mathbb{E}\left[\sum_{h' \geq h} r_{h'}^i\right] - Q_h^*(s_h^i, a_h^i) - \mathbb{E}[b_i] = Q^{\tilde{\pi}_h}(s_h^i, a_h^i) - Q^*(s_h^i, a_h^i) + \left(Q^*(s_h^i, a_h^i) - Q^{\tilde{\pi}_h}(s_h^i, a_h^i)\right) = 0.$$

By (9), $\Pr[b_i \neq 0] \leq \epsilon_0$. Thus by Lemma 14,

$$\mathbb{E}_{\pi \sim \text{Unif}(\Pi_h), s_h \sim \mathcal{D}_h^\pi, a_h \sim \rho_{s_h}} \left[\left|\phi(s_h, a_h)^\top (\theta_h - \theta_h^*)\right|^2\right] \leq 8\left(\epsilon_1 + \lambda_{\text{ridge}}\right) + 288\epsilon_0^{1.5} C_{\text{hyper}}^{2.5} d^{4.5} \left(\frac{2B}{\epsilon}\right)^{0.5}$$

$$\leq 16\epsilon^3 + 288\epsilon^{2.5} C_{\text{hyper}}^{2.5} d^{4.5} (2B)^{0.5}.$$

It follows that $\forall \pi \in \Pi_h$,

$$\mathbb{E}_{s_h \sim \mathcal{D}_h^\pi, a_h \sim \rho_{s_h}} \left[\left|\phi(s_h, a_h)^\top (\theta_h - \theta_h^*)\right|^2\right] \leq |\Pi_h| \cdot \left(16\epsilon^2 + 288\epsilon^{2.5} C_{\text{hyper}}^{2.5} d^{4.5} (2B)^{0.5}\right) \leq \frac{\Delta_{\min}^2 \epsilon_0}{120 H B d^2},$$

where we used the fact $\epsilon \leq \text{poly}(\Delta_{\min}, 1/C_{\text{hyper}}, 1/d, 1/H)$. Now, by Lemma 5,

$$\mathbb{E}_{s_h \sim \mathcal{D}_h^\pi} \left[\sup_{a \in \mathcal{A}} \left|\phi(s_h, a)^\top (\theta_h - \theta_h^*)\right|^2\right] \leq \mathbb{E}_{s_h \sim \mathcal{D}_h^\pi} \left[\sup_{a \in \mathcal{A}} \|\phi(s_h, a)\|_{(\Sigma_h^\pi)^{-1}}^2\right] \cdot \|\phi_h - \phi_h^*\|_{\Sigma_h^\pi}^2$$

$$\leq d^2 \cdot \frac{\Delta_{\min}^2 \epsilon_0}{120 H B d^2} \leq \frac{\Delta_{\min}^2 \epsilon_0}{4H}.$$

This proves the second claim.

**Proof of (7).** Now, let us prove the first claim, assuming that the second claim holds for the last update of any $\theta_h$. Consider Line 9 when checking for $\pi \in \Pi_h$ for layer $h'$. Recall that

$$\Sigma_{h'}^{\tilde{\pi}_h} = \mathbb{E}_{s_{h'} \sim \mathcal{D}_h^{\tilde{\pi}_h}, a_{h'} \sim \rho_{s_{h'}}} \left[\phi(s_{h'}, a_{h'}) \phi(s_{h'}, a_{h'})^\top\right].$$

Similar to the proof under Assumption 3, we can bound $\|\theta_{h'} - \theta_{h'}^*\|_{\Sigma_{h'}^{\tilde{\pi}_h}}$ by

$$\|\theta_{h'} - \theta_{h'}^*\|_{\Sigma_{h'}^{\tilde{\pi}_h}}^2 \leq \|\theta_{h'} - \theta_{h'}^*\|_{\Sigma_{h'}^*}^2 \cdot \|(\Sigma_{h'}^*)^{-\frac{1}{2}} \Sigma_{h'}^{\tilde{\pi}_h} (\Sigma_{h'}^*)^{-\frac{1}{2}}\|_2$$

$$\leq 3\beta B \cdot \left(\lambda_r \cdot \left(\frac{2}{\lambda_{\text{ridge}}}\right)^2 + \frac{\Delta_{\min}^2 \epsilon_0}{120 H B d^2}\right)$$

$$\leq 96 B \epsilon^3 + \frac{\Delta_{\min} \epsilon_0}{5 H d^2}.$$

By Lemma 5, $\mathbb{E}_{s_{h'}\sim\mathcal{D}_{h'}^{\tilde{\pi}_h}}\left[\sup_{a\in\mathcal{A}}\|\phi(s_{h'},a)\|_{(\Sigma_{h'}^{\tilde{\pi}_h})^{-1}}^2\right]\le d^2$. Consequently

$$\mathbb{E}_{s_{h'}\sim\mathcal{D}_{h'}^{\tilde{\pi}_h}}\left[\sup_{a\in\mathcal{A}}\left|\phi(s_{h'},a)^\top(\theta_{h'}-\theta_{h'}^*)\right|^2\right]\le\mathbb{E}_{s_{h'}\sim\mathcal{D}_{h'}^{\tilde{\pi}_h}}\left[\|\theta_{h'}-\theta_{h'}^*\|_{\Sigma_{h'}^{\tilde{\pi}_h}}^2\cdot\sup_{a\in\mathcal{A}}\|\phi(s_{h'},a)\|_{(\Sigma_{h'}^{\tilde{\pi}_h})^{-1}}^2\right]$$

$$\le 96B\epsilon^3 d^2+\frac{\Delta_{\min}\epsilon_0}{5H}\le\frac{\Delta_{\min}\epsilon_0}{4H}.$$

In the last inequality we used $\epsilon_0=\epsilon^2$ and $\epsilon\le\mathrm{poly}(\Delta_{\min},1/d,1/H)$. This proves (7). Finally the failure probability is controlled in Lemma 8. $\qquad\square$

**Lemma 6** (Covariance concentration [Tropp, 2015]). *Suppose $M_1,\cdots,M_N\in\mathbb{R}^{d\times d}$ are i.i.d. random matrices drawn from a distribution $\mathcal{D}$ over positive semi-definite matrices. If $\|M_t\|_F\le 1$ almost surely and $N=\Omega\left(\frac{d\log(d/\delta)}{\epsilon^2}\right)$, then with probability $1-\delta$,*

$$\left\|\frac{1}{N}\sum_{i=1}^N M_t-\mathbb{E}_{M\sim\mathcal{D}}[M]\right\|_2\le\epsilon.$$

**Lemma 7** (Risk bound for ridge regression, Lemma A.2 Du et al. [2019]). *Suppose that $(x_1,y_1),\cdots,(x_N,y_N)$ are i.i.d. data drawn from $\mathcal{D}$ with*

$$y_i=\theta^\top x_i+b_i+\xi_i,$$

*where $\mathbb{E}_{(x_i,y_i)\sim\mathcal{D}}[b_i^2]\le\eta$, $|\xi_i|\le 2n$ almost surely and $\mathbb{E}[\xi_i]=0$. Let the ridge regression estimator be*

$$\hat{\theta}=\left(\sum_{i=1}^N x_i x_i^\top+N\lambda_{\mathrm{ridge}}\cdot I\right)^{-1}\cdot\sum_{i=1}^N x_i y_i.$$

*If $N=\Omega\left(\frac{d}{\epsilon_N^2}\log(\frac{d}{\delta})\right)$, then with probability at least $1-\delta$,*

$$\mathbb{E}_{x\sim\mathcal{D}}\left[\left((\hat{\theta}-\theta)^\top x\right)^2\right]\le 4\left(\eta+\epsilon_N+\lambda_{\mathrm{ridge}}\right).$$

**Lemma 8** (Failure probability). *Define the following events regarding the execution of Algorithm 1.*

1. *$\Omega_1$: Each time $\Sigma_h$ is updated,*

$$\left\|\Sigma_h-\mathbb{E}_{\pi\sim Unif(\Pi_h),s_h\sim\mathcal{D}_h^\pi,a_h\sim\rho_{s_h}}\left[\phi(s_h,a_h)\phi(s_h,a_h)^\top\right]\right\|_2\le\epsilon_2.\tag{10}$$

2. *$\Omega_2$: Each time $\theta_h$ is updated,*

$$\mathbb{E}_{\pi\sim Unif(\Pi_h),s\sim\mathcal{D}_h^\pi,a\sim\rho_s}\left[\left((\theta_h-\theta_h^*)^\top\phi(s,a)\right)^2\right]\le 4\left(\eta+\epsilon_1+\lambda_{\mathrm{ridge}}\right),\tag{11}$$

   *where $\eta$ is defined as in Lemma 7.*

3. *$\Omega_3$: Each time $\theta_h$ is updated,*

$$\mathbb{E}_{\pi\sim Unif(\Pi_h),s\sim\mathcal{D}_h^\pi,a\sim\rho_s}\left[\left((\theta_h-\theta_h^*)^\top\phi(s,a)\right)^2\right]\le 288\eta^{1.5}C^{2.5}d^{4.5}\left(\frac{2B}{\epsilon}\right)^{0.5},\tag{12}$$

   *where $\eta$ and $C$ are defined as in Lemma 14.*

*Then under Assumption 3, $\Pr[\Omega_1\cap\Omega_2]\ge 1-\epsilon$. Alternatively, under Assumption 4, $\Pr[\Omega_1\cap\Omega_3]\ge 1-\epsilon$.*

*Proof.* Note that $N\ge\frac{d\log(1/\epsilon_2)}{\epsilon_2^2}$ where $\epsilon_2\le\frac{\epsilon^6}{d}$. Therefore, by Lemma 6, each time $\Sigma_h$ is updated, (10) holds with probability at least $1-\epsilon^2$.

As for (11), note that $N\ge\frac{d\log(1/\epsilon_2)}{\epsilon_2^2}\gg\frac{d}{\epsilon_1^2}\cdot\log(\frac{d}{\epsilon^2})$. Thus by Lemma 7, each time $\theta_h$ is updated, (11) holds with probability at least $1-\epsilon^2$.

Similarly, for (12), under the choice of parameters under Assumption 4, $N \geq \frac{d}{\epsilon_2^3} \gg \left(\frac{d}{\epsilon_2^2} + \frac{1}{\eta}\right) \ln \frac{2dB}{\epsilon} + \frac{2B}{\epsilon}$. Thus by Lemma 14, the probability that (12) is violated each step is at most $\epsilon/2B$.

Note that when the algorithm terminates, the $\Sigma_h$ and $\theta_h$ are updated at most $|\Pi_h|$ times. Also note that, if during the first $B$ updates, neither (10) nor (11) are violated, by Lemma 10 it follows that $|\Pi_h| \leq B$ when the algorithm terminates. In other words,

$$\Pr[\Omega_1 \cup \Omega_2] \geq 1 - B \cdot 2\epsilon^2 \geq 1 - \epsilon.$$

Similarly, under Assumption 4,

$$\Pr[\Omega_1 \cup \Omega_3] \geq 1 - B \cdot \epsilon^2 - B \cdot \frac{\epsilon}{2B} \geq 1 - \epsilon.$$

$\square$

**Lemma 9** (Distribution shift error checking). *Assume that $\epsilon_2 < \min\{\frac{1}{2}\beta\lambda_r, \frac{\lambda_r}{2B}\}$. Consider the `if` clause when checking for $\pi_h \in \Pi_h$, i.e. when computing $\|\Sigma_{h'}^{-\frac{1}{2}}\hat{\Sigma}_{h'}\Sigma_{h'}^{-\frac{1}{2}}\|_2$. Define*

$$M_1 := \frac{\lambda_r}{|\Pi_{h'}|}I + \mathbb{E}_{\pi\sim Unif(\Pi_{h'}), s_{h'}\sim\mathcal{D}_{h'}^{\pi}, a_{h'}\sim\rho_{s_{h'}}}\left[\phi(s_{h'}, a_{h'})\phi(s_{h'}, a_{h'})^\top\right],$$

*and*

$$M_2 := \mathbb{E}_{s_{h'}\sim\mathcal{D}_{h'}^{\tilde{\pi}_h}, a_{h'}\sim\rho_{s_{h'}}}\left[\phi(s_{h'}, a_{h'})\phi(s_{h'}, a_{h'})^\top\right].$$

*Then under the event $\Omega$ defined in Lemma 8, when $\|\Sigma_{h'}^{-\frac{1}{2}}\hat{\Sigma}_{h'}\Sigma_{h'}^{-\frac{1}{2}}\|_2 \leq \beta|\Pi_{h'}|$,*

$$\|M_1^{-1/2}M_2M_1^{-1/2}\|_2 \leq 3\beta|\Pi_{h'}|.$$

*When $\|\Sigma_{h'}^{-\frac{1}{2}}\hat{\Sigma}_{h'}\Sigma_{h'}^{-\frac{1}{2}}\|_2 \geq \beta|\Pi_{h'}|$,*

$$\|M_1^{-1/2}M_2M_1^{-1/2}\|_2 \geq \frac{1}{4}\beta|\Pi_{h'}|.$$

*Proof.* By Lemma 6,

$$\|M_1 - \Sigma_{h'}\|_2 \leq \epsilon_2 \leq \frac{\lambda_r}{2B} \leq \frac{1}{2}\lambda_{\min}(\Sigma_{h'}).$$

Thus $\frac{1}{2}\Sigma_{h'} \preccurlyeq M_1 \preccurlyeq 2\Sigma_{h'}$. Also by Lemma 6, $\|M_2 - \hat{\Sigma}_{h'}\|_2 \leq \epsilon_2$. Therefore, if $\|\Sigma_{h'}^{-\frac{1}{2}}\hat{\Sigma}_{h'}\Sigma_{h'}^{-\frac{1}{2}}\|_2 \geq \beta|\Pi_{h'}|$,

$$\|M_1^{-1/2}M_2M_1^{-1/2}\|_2 \geq \frac{1}{2}\|\Sigma_{h'}^{-1/2}M_2\Sigma_{h'}^{-1/2}\|_2 \geq \frac{1}{2}\|\Sigma_{h'}^{-1/2}\hat{\Sigma}_{h'}\Sigma_{h'}^{-1/2}\|_2 - \frac{1}{2}\epsilon_2\|\Sigma_{h'}^{-1}\|_2$$

$$\geq \frac{1}{2}\beta|\Pi_{h'}| - \frac{1}{2}\epsilon_2 \cdot \frac{|\Pi_{h'}|}{\lambda_r} \geq \frac{1}{4}\beta|\Pi_{h'}|.$$

Similarly, when $\|\Sigma_{h'}^{-\frac{1}{2}}\hat{\Sigma}_{h'}\Sigma_{h'}^{-\frac{1}{2}}\|_2 \leq \beta|\Pi_{h'}|$,

$$\|M_1^{-1/2}M_2M_1^{-1/2}\|_2 \leq 2\|\Sigma_{h'}^{-1/2}M_2\Sigma_{h'}^{-1/2}\|_2 \leq 2\|\Sigma_{h'}^{-1/2}\hat{\Sigma}_{h'}\Sigma_{h'}^{-1/2}\|_2 + 2\epsilon_2\|\Sigma_{h'}^{-1}\|_2$$

$$\leq 2\beta|\Pi_{h'}| + 2\epsilon_2 \cdot \frac{|\Pi_{h'}|}{\lambda_r} \leq 3\beta|\Pi_{h'}|.$$

$\square$

**Lemma 10** (Lemma A.6 in Du et al. [2019]). *Under the event $\Omega_1$ defined in Lemma 8, $|\Pi_h| \leq B$ for all $h \in [H]$.*

*Proof.* We provide a proof for completeness. Fix a level $h' \in [H]$. Define

$$A := \lambda_r I + \sum_{\pi\in\Pi_{h'}} \mathbb{E}_{s_{h'}\sim\mathcal{D}_{h'}^{\tilde{\pi}_h}, a_{h'}\sim\rho_{s_{h'}}}\left[\phi(s_{h'}, a_{h'})\phi(s_{h'}, a_{h'})^\top\right].$$

By the update rule at Line 6, $|\Pi_{h'}|$ is expanded if and only if the `if` clause at Line 5 returns False when checking for some $\tilde{\pi}_h$. By Lemma 9, define

$$M := \mathbb{E}_{s_{h'} \sim \mathcal{D}_{h'}^{\tilde{\pi}_h}, a_{h'} \sim \rho_{s_{h'}}} \left[ \phi(s_{h'}, a_{h'}) \phi(s_{h'}, a_{h'})^\top \right],$$

then

$$\|A^{-1/2} M A^{-1/2}\|_2 \geq \frac{1}{4} \beta = 2.$$

Note that after $\Pi_{h'}$ is updated to $\Pi_{h'} \cup \{\tilde{\pi}_h\}$, $A$ would be updated to $A + M$. Observe that

$$\det(A + M) = \det(A) \cdot \det\left(I + A^{-1/2} M A^{-1/2}\right) \geq 3 \det(A).$$

Therefore during the execution of the algorithm,

$$\det(A) \geq 3^{|\Pi_{h'}|} \cdot \lambda_r^d.$$

On the other hand, since $\|\phi(s, a)\phi(s, a)^\top\|_2 \leq 1$,

$$\det(A) \leq (\lambda_r + |\Pi_{h'}|)^d.$$

The lemma follows by solving $3^{|\Pi_{h'}|} \cdot \lambda_r^d \leq (\lambda_r + |\Pi_{h'}|)^d$. $\qquad\square$

## 6 Analysis of Ridge Regression under Hypercontractivity

Recall that a distribution $\mathcal{D}$ is $(C, 4)$-hypercontractive if $\forall v$,

$$\mathbb{E}_{x \sim \mathcal{D}}[(x^\top v)^4] \leq C \cdot \left(\mathbb{E}_{x \sim \mathcal{D}}[(x^\top v)^2]\right)^2.$$

In this section we prove an strengthened version of Lemma 7 for hypercontractive distributions (Lemma 14), which may be of independent interest.

**Lemma 11.** *Let $x$ be a $d$-dimensional r.v. If the distribution of $x$ is $(C, 4)$-hypercontractive and isotropic (i.e. $\mathbb{E}[xx^\top] = I$), then*

$$\Pr[\|x\|_2 > t] \leq \frac{Cd^2}{t^4}.$$

*Proof.* Consider a Gaussian random vector $v \sim N(0, I)$. Then

$$\mathbb{E}_v[(x^\top v)^4] = \|x\|^4 \cdot \mathbb{E}_{\xi \sim N(0,1)} \xi^4 = 3\|x\|^4.$$

Therefore

$$\mathbb{E}_x[\|x\|^4] = \frac{1}{3} \mathbb{E}_{x,v}[(x^\top v)^4] \leq \frac{C}{3} \mathbb{E}_v \left(\mathbb{E}_x(x^\top v)^2\right)^2$$

$$\leq \frac{C}{3} \mathbb{E}_v \|v\|^4 = \frac{C \cdot (d^2 + 2d)}{3} \leq d^2 C.$$

The claim then follows from Markov's inequality. $\qquad\square$

**Lemma 12.** *If the $x_1, \cdots, x_n$ are i.i.d. samples from a $(C, 4)$-hypercontractive distribution. Let $\sigma(\cdot)$ denote the decreasing order of $\|x_i\|_2$. Then with probability $1 - \delta$,*

$$\sum_{k=1}^m \|x_{\sigma(k)}\|_2 = 3\delta^{-1/4} n^{1/4} m^{3/4} C^{1/4} d^{1/2}.$$

*Proof.* Fix $k \in [m]$. Set $t = \alpha \left(\frac{Cd^2 n}{k}\right)^{1/4}$. By Lemma 11,

$$\Pr[\|x_{\sigma(k)}\|_2 > t] \leq \binom{n}{k} \Pr[\|x\| > t]^k \leq \binom{n}{k} \cdot \left(\frac{Cd^2}{t^4}\right)^k$$

$$\leq \frac{n^k}{k!} \cdot \frac{k^k}{\alpha^{4k} n^k} \leq \left(\frac{e}{\alpha^4}\right)^k.$$

Choosing $\alpha = \left(\frac{2e}{\delta}\right)^{1/4}$ gives $\Pr[\|x_{\sigma(k)}\|_2 > t] \le (\delta/2)^k$. By a union bound, with probability $1 - \delta$,

$$\sum_{i=1}^{m} \|x_{\sigma(i)}\|_2 \le \sum_{k=1}^{m} (2e/\delta)^{1/4} \left(\frac{Cd^2 n}{k}\right)^{1/4} \le 3\delta^{-1/4} n^{1/4} m^{3/4} C^{1/4} d^{1/2}.$$

$\square$

**Lemma 13** (Lemma 3.4 Bakshi and Prasad [2020]). *If $\mathcal{D}$ is $(C, 4)$-hypercontractive and $x_1, \cdots, x_n$ are i.i.d. samples drawn from $\mathcal{D}$. Let $\Sigma := \mathbb{E}_{x \sim \mathcal{D}}[xx^\top]$. With probability $1 - \delta$,*

$$\left(1 - \frac{Cd^2}{\sqrt{n\delta}}\right) \Sigma \preccurlyeq \frac{1}{n} \sum_{i=1}^{n} x_i x_i^\top \preccurlyeq \left(1 + \frac{Cd^2}{\sqrt{n\delta}}\right) \Sigma.$$

**Lemma 14** (Risk bound for ridge regression with hypercontractivity). *Suppose that $(x_1, y_1), \cdots, (x_N, y_N)$ are i.i.d. data drawn from $\mathcal{D}$ with*

$$y_i = \theta^\top x_i + b_i + \xi_i,$$

*where $\Pr[b_i \ne 0] \le \eta$, $\|b\|_\infty \le 1$, $|\xi_i| \le 1$, and $\mathbb{E}[\xi_i] = 0$. Assume that distribution of $x$ is $(C, 4)$-hypercontractive (see Assumption 4). Let the ridge regression estimator be*

$$\hat{\theta} = \left(\sum_{i=1}^{N} x_i x_i^\top + N\lambda_{\mathrm{ridge}} \cdot I\right)^{-1} \cdot \sum_{i=1}^{N} x_i y_i.$$

*If $N = \Omega\left(\left(\frac{d}{\epsilon_N^2} + \frac{1}{\eta}\right)\log(\frac{d}{\delta}) + \frac{1}{\delta}\right)$, then with probability at least $1 - \delta$,*

$$\mathbb{E}_{x \sim \mathcal{D}}\left[\left((\hat{\theta} - \theta)^\top x\right)^2\right] \le 8\left(\epsilon_N + \lambda_{\mathrm{ridge}}\right) + 288\eta^{1.5} C^{2.5} d^{4.5} \delta^{-0.5}.$$

*Proof.* Define $\hat{\Sigma} := \frac{1}{N} \sum_{i=1}^{N} x_i x_i^\top$ and $\Sigma := \mathbb{E}_{x \sim \mathcal{D}}[xx^\top]$. Then

$$\hat{\theta} = \frac{1}{N} \left(\lambda_{\mathrm{ridge}} I + \hat{\Sigma}\right)^{-1} \sum_{i=1}^{N} \left(x_i x_i^\top \theta + x_i \cdot \xi_i + x_i \cdot b_i\right)$$

$$= \underbrace{\frac{1}{N} \left(\lambda_{\mathrm{ridge}} I + \hat{\Sigma}\right)^{-1} \sum_{i=1}^{N} b_i x_i}_{(a)} + \underbrace{\frac{1}{N} \left(\lambda_{\mathrm{ridge}} I + \hat{\Sigma}\right)^{-1} \sum_{i=1}^{N} \left(x_i x_i^\top \theta + x_i \cdot \xi_i\right)}_{(b)}.$$

By Lemma 7, $\|\theta - (b)\|_\Sigma^2 \le 4(\epsilon_N + \lambda_{\mathrm{ridge}})$. It remains to bound the $\|\cdot\|_\Sigma$ norm of $(a)$.

First, by Hoeffding's inequality, with probability $1 - \delta$, $\|b\|_0 = \sum_{i=1}^{n} I[b_i \ne 0] \le 2\eta N$. Define $z_i := \Sigma^{-1/2} x_i$ to be the normalized input. It can be seen that $\mathbb{E}[z_i z_i^\top] = I$ and that the distribution of $z_i$ is also hypercontractive. By Lemma 12, with probability $1 - 2\delta$,

$$\sum_{i=1}^{n} \|z_i\|_2 \cdot I[b_i \ne 0] \le 3\delta^{-1/4} N^{1/4} (2\eta N)^{3/4} (Cd^2)^{1/4}.$$

It follows that with probability $1 - 2\delta$,

$$\|(a)\|_\Sigma = \frac{1}{N} \left\|\hat{\Sigma}^{-1} \sum_{i=1}^{N} x_i b_i\right\|_\Sigma \le \frac{1}{N} \sum_{i=1}^{N} \|\Sigma^{1/2} \hat{\Sigma}^{-1} x_i b_i\|_2$$

$$= \frac{1}{N} \sum_{i=1}^{N} \|\Sigma^{1/2} \hat{\Sigma}^{-1} \Sigma^{1/2} z_i b_i\|_2$$

$$\le \frac{1}{N} \|\Sigma^{1/2} \hat{\Sigma}^{-1} \Sigma^{1/2}\|_2 \cdot \sum_{i=1}^{N} \|z_i\|_2 \cdot H \cdot I[b_i \ne 0]$$

$$\le 3H \left(1 + \frac{Cd^2}{\sqrt{N\delta}}\right) \cdot \delta^{-1/4} N^{-3/4} (2\eta N)^{3/4} (Cd^2)^{1/4}$$

$$\le 12H\eta^{0.75} \cdot C^{\frac{5}{4}} d^{\frac{9}{4}} \delta^{-\frac{1}{4}}.$$

Therefore

$$\|\hat{\theta} - \theta\|_\Sigma^2 \leq 2\|\hat{\theta} - (b)\|_\Sigma^2 + 2\|(a)\|_\Sigma^2$$
$$\leq 8(\epsilon_N + \lambda_{\text{ridge}}) + 288\eta^{1.5}C^{2.5}d^{4.5}\delta^{-0.5}.$$

$\square$