# OpenReview forum: "An Exponential Lower Bound for Linearly Realizable MDP with Constant Suboptimality Gap"
_NeurIPS.cc/2021/Conference — NeurIPS 2021 Oral_

### Official Review · Reviewer_vgYe · 2021-07-06

**Rating:** 8
**Confidence:** 4

**Summary:**

This is a technical paper that considers online RL with linear function approximation under the assumptions: 1) Q^* is realizable, 2) gap = V^*(s) - Q^*(s,a) > 0 for all (s,a). The first result is an unfortunate impossibility result that gives an exponential sample lower bound without further assumptions. Then under additional assumptions on system dynamics, a variation of existing algorithm (Difference Maximization Q-Learning) returns a near-optimal policy in polynomial time.

**Limitations And Societal Impact:**

The limitations and remarks are well pointed out in the discussion section.

**Main Review:**

The paper considers an interesting open question on the online RL with linear function approximation. Realizability and constant minimum gap are commonly used, and reasonable assumptions for this setting, but nevertheless, the paper shows that it is still a hard problem without further assumptions. This negative result is a good addition to the current literature. For the upper bound, a major contribution is to remove the linear dependency on the number of actions with a nice adaptation of G-optimal experimental design. Overall, the paper is very well-written and presents solid improvements/contributions to the RL theory, hence I recommend the acceptance.




Main Comments

- It seems that there is a caveat in this negative result: there is no limit on the action space and thus there could be exponentially many possible actions to choose. It would have been better clarified if the same negative result can be obtained when the number of actions are finite (like in many practical problems).

- Could you explain what Assumption 4 (hypercontractivity) means in the RL world more explicitly? Unlike in robust-statistics where the hypercontractivity is crucial for the success of specific moment-based methods, for the proposed algorithm it is hard to see why it helps to overcome the impossibility result. Is Assumption 4 some sort of uniform ergodicity? or is it some variation of (nearly) deterministic systems?

- How broadly Assumption 3 or 4 can be applied in practice? What are some practical examples whose transition kernels are not completely deterministic but still satisfying these assumptions?


Minor Comments

- Line 225 - 226: One style of assumption ... [Zanette et al., 2020] -> I don't understand how this work is relevant to your assumptions.

- Section 5.1: It might be nicer to state and emphasize what is the major challenge / target to improve at the beginning of the section.

**Time Spent Reviewing:**

4

---

> ### Author Response · Authors · 2021-08-11
> **Response**
>
> We would like to thank the reviewer for the insightful and positive comments.
>
>
> We agree that our hard instance requires an exponential number of actions to work, and that will be made clearer in the next version. However, it is unclear to us whether the exponential lower bound still holds when the number of actions is polynomially small. In fact, we spent a serious amount of time on this problem and got no luck. We also tried hard instances similar to that suggested by the reviewer and did not succeed. Given this, it might still be possible that the problem is solvable with a polynomial number of samples when the action space has constant size (or even $\mathrm{poly}(d)$ size).
>
> Intuitively, hypercontractivity states that for any policy, features of the induced state-action distribution satisfy certain anti-concentration properties. Since Gaussian distributions (with arbitrary covariance matrices) satisfy hypercontractivity, a practical example where hypercontractivity holds is the case when the state transition has Gaussian noise (as common in control settings). We will add more discussion in the next version.

---

### Official Review · Reviewer_ca1d · 2021-07-11

**Rating:** 8
**Confidence:** 3

**Summary:**

This paper studies lower bounds for the linear-$Q^*$ setting with a gap>0 assumption. Recently Weisz et al prove a milestone result that the linear-$Q^*$ setting requires exponential in d sample complexity. However the gap in these instances, the gap is exponentially small too. It is known that with linear-$Q^*$ assumption + gap>0 , an optimal policy can be learned with polynomial (in all parameters) sample complexity given access to a generative model. The result of this paper thus establishes a separation in the sample complexity compared to the online setting under the same assumptions.
In addition, the authors also propose algorithms that break the exponential sample complexity lower bounds in this setting under either of (I) a low variance assumption on the variance of the value of any policy from $V^*$ under its own state distribution, or (II) a $(C,4)$-hypercontractivity assumption on the feature vector distribution under any policy. The proof under the latter assumption follows from a novel analysis of least squares regression under hypercontractivity assumptions.

**Ethical Concerns:**

None to the best of my knowledge.

**Limitations And Societal Impact:**

Yes, the authors have addressed the limitations / future work and also potential negative impacts of their work sufficiently.

**Main Review:**

Overall I think the contribution of the paper is significant and the presentation is quite clear. To the best of my knowledge most of the related works have been sufficiently addressed.
Here are my suggestions for the paper:

* In the paper, it would help (subject to space constraints) to (i) include a figure detailing a rough description of the lower bound in the paper and (ii) a discussion comparing the current construction to the previous lower bound of Weisz et al.

* In section 4.1 it would help to add a line that the total number of states in the MDP is also equal to $m$.

* In the presentation, I sometimes got confused with notations such as $\bar{a}_1$, $\bar{a}_2$, etc. which are in fact used to refer to states and not actions. This is a bit jarring at times. I think in general it may help to only index the states by $\bar{i}$ and $\bar{j}$ and avoid using the notation $a_\cdot$ entirely here.

* In Theorem 2, it might helpful for a reader to see a more explicit description of the number of trajectories consumed by the algorithm instead of $\text{poly} (1/\epsilon)$.

* In the case the action space constrained to be $\text{poly}(d)$, is there any conjectured approach for a lower bound?
A natural approach is to split each state in the current (exponential action space) MDP into an exponential number of states with a constant number of actions each, and inducing the uniform distribution over these states upon taking any action in the previous state. However, this approach fails because even the value of the optimal policy becomes exponentially small. But of course, there is a lot more room to work around here. So, I wonder: do you believe the flexibility of the next-state distribution here is sufficient to get such a lower bound instance to work when the action space is only polynomially large?


**Time Spent Reviewing:**

5

---

> ### Author Response · Authors · 2021-08-11
> **Response**
>
> We would like to thank the reviewer for the insightful and positive comments.
>
> Suggestions on the writing will be taken. In particular, we will add a figure to describe the high-level construction of our hard instance, together with a new paragraph comparing our construction to that of Weisz et al..
>
> It is unclear to us whether the exponential lower bound still holds when the number of actions is polynomially small. In fact, we spent a serious amount of time on this problem and got no luck. We also tried hard instances similar to that suggested by the reviewer and did not succeed. Given this, it might still be possible that the problem is solvable with a polynomial number of samples when the action space has constant size (or even $\mathrm{poly}(d)$ size).

---

### Official Review · Reviewer_HMEJ · 2021-07-16

**Rating:** 8
**Confidence:** 4

**Summary:**

The authors present exponential lower bounds under perfect realizability when the model must be learned online even with constant action-value function gaps. An upper bound is presented under a new hypercontractivity assumption.

**Main Review:**

I thank the authors for their submission, I liked the work very much.

Strengths:
+ the work has some genuinely new ideas in the construction (although it leverages prior techniques like JL lemma)
+ it clarifies what’s achievable in the far more interesting online setting (i.e., the separation with the generative model setting)
+ the construction is easier than Weisz et al ’20, making it more amenable to explanation

I read the upper bound section a bit faster, but I’d like the authors to clarify a bit better the assumption about hypercontractivity  (in particular, its meaning in the RL setting) in the final version of the work.

The work makes a non-trivial technical contribution, but most importantly, it presents a key new result: while Weisz et al ’20 also give MDPs that are hard to learn in the online setting considered here, the construction there is more contrived / pathological (this is by necessity). By making the setting more realistic (i.e., online), learning becomes harder and the authors leverage a less pathological construction to still get ~ e^d complexity, further supporting the idea that such hardness is more ``   `realistic'.

 I support the work for publication to NeurIPS 2021.


**Time Spent Reviewing:**

3

---

> ### Author Response · Authors · 2021-08-11
> **Response**
>
> We would like to thank the reviewer for the insightful and positive comments.
>
> Intuitively, hypercontractivity states that for any policy, features of the induced state-action distribution satisfy certain anti-concentration properties. Since Gaussian distributions (with arbitrary covariance matrices) satisfy hypercontractivity, a practical example where hypercontractivity holds is the case when the state transition has Gaussian noise (as common in control settings). We will add more discussion in the next version.

---

### Official Review · Reviewer_PcYF · 2021-07-19

**Rating:** 8
**Confidence:** 4

**Summary:**

This paper considers online RL with linear realizability of Q* and a suboptimality gap assumption. It shows an exponential sample complexity lower bound holds for this setting, but under an additional assumption of either (a) or (b), there is an algorithm (modified DMQ) that achieves a polynomial upper bound, showing sample-efficient RL is possible in this setting.
The additional assumption required for this is either (a) low variance of value differences between some policy pi and the optimal policy, or (b) hypercontractivity of the distribution of features encountered along any policy.

The negative result is a modification of Weisz et al 2020’s construction of hard MDPs into a “leaky” graph that allows for a large suboptimality gap.
The positive result is similar in algorithm and analysis to Du et al 2019b, with the novelties that (1) a number-of-actions factor in the sample complexity is avoided by sampling actions from an optimal design, and (2) the method is shown to work for assumption (b).

(1) is important to make the separation between the lower and upper bounds clear as the lower bound uses exponentially many actions.
As the authors point out, these results also imply that there is an exponential separation between online RL and RL with a generative model, as in the latter, sample-efficient learning is possible in the same setting as this paper’s lower bound.

**Limitations And Societal Impact:**

yes

**Main Review:**

LOWER BOUND:

The construction for the bound seems to be largely inspired by Weisz et al. 2020’s construction: the crucial difference to achieve the suboptimality gap is to construct a "leaky” graph instead of downscaling the features. This way, the downscaling is essentially achieved by the transitions. The two techniques allow for a similar information theoretic argument to finish the bound. The result is both stronger and weaker: the suboptimality gap is a stronger assumption, but the lower bound only works in the online RL setting (as opposed to the generative model).
I was happy to see a hard MDP construction similar to Weisz et al 2020’s presented (in my opinion) much more cleanly.

- I am missing the definition of Pr_M at appendix line 61.
- I am unsure why the appendix "addressing footnote 3" is necessary: could we not just bijectively remap the m-1 actions of each state to [m-1]?

UPPER BOUND:

I would like to see some assumptions clarified here, as there might be inaccuracies. Below is my best attempt at understanding them. Whether I am right or wrong here, the effort required to address my issues in the paper should be fairly small.
- First, in line 147, I believe the precise assumption on theta* and phi's 2-norms is that they are <=1 (instead of O(1); see eg appendix line 189). I would like to see this stated as a separate assumption. The trouble with the argument that this "can always be achieved via rescaling" is that one would have to rescale the rewards, and consequently epsilon too, to achieve the same guarantee. This means that the final bound will scale polynomially with these 2-norms, instead of (what might be possible) logarithmically. I believe for this reason the assumption should be made explicit, as it is in Du et al 2019b. It would be nice for this assumption to be referenced in the proof too, (eg appendix lines 103, 189, lemma 7, etc.).
- Second, in lemma 7, Du et al 2019b’s relevant lemma is A.3 instead of A.2, (at least in the versions available to me), and it is stated slightly differently. I would like to see these differences proved or explained. These are:
  - a condition on eta<=1
  - |\xi|<=1 (I'm not sure what n refers to in appendix line 149)
  - eta^2 in the bound instead of eta
- Third, Du et al 2019b uses an additional assumption, that rewards are nonnegative and the sum of rewards on *any* trajectory is bounded by 1. I believe this assumption is missing from this paper, but seems to be implicitly used at places like (other than possibly at lemma 7): appendix line 98 (for the value bound), and line 107 (to bound \xi almost surely; here this bound would be <=1, not sure where the 2 comes from). Both of these arguments feel somewhat unfinished to me.

For the low variance and hypercontractivity assumptions introduced at the beginning of section 5, it would be nice to see examples or descriptions of the kinds of MDPs whose value functions satisfy this assumption. The current description focuses on the value functions, not on the MDPs that might induce such value functions. Having this could help interpreting the generality of these assumptions (and hence the corresponding upper bound too).

Does the upper bound work under a near-realizability assumption (eg. ||Q*-phi^T theta*||_\infty <= something *very* small)?

- Line 315: is known -> was known
- Line 7: such a lower bound (add bound)
- Appendix line 119: consider better justifying line 3 to line 4, bound of theta-theta* by including the calculation

**Time Spent Reviewing:**

16

---

> ### Author Response · Authors · 2021-08-11
> **Response**
>
> We would like to thank the reviewer for carefully reading our submission and providing many insightful comments.
>
> 1. Here we used $\Pr_M[s' | s, a]$ to denote the probability of transiting from $(s, a)$ to $s'$ in MDP $M$. We will change the notation to $P_M(s' \mid s, a)$ in the next version to make the notation consistent.
>
> 2. Indeed remapping the actions will result in a much simpler proof. We will adopt such proof in the next version. Thanks for the suggestion.
>
> 3. We agree that the precise assumption on theta* and phi's 2-norms is that they are <=1, and that will be made clear in the next version.
>
> 4. The condition that $\eta \le 1$ was missed in the statement of Lemma 7 (note that the error bound was correctly stated since in our case $\eta = \overline{b}^2$). The correct constraint on $\xi$ should be $|\xi| \le 2$. Note that this will only increase the sample complexity by a constant factor which was hidden in the $\Omega$ notation. These typos will be fixed in the next version.
>
> 5. We did assume that rewards are nonnegative and the sum of rewards on any trajectory is bounded by 1, and that will be made clear in the next version. We will also connect relevant parts in the proof to this assumption for clarity.
>
> We note that hypercontractivity is an assumption on the distribution of the features instead of the value functions. Intuitively, hypercontractivity states that for any policy, the features of the induced state-action distribution satisfy certain anti-concentration properties. Since Gaussian distributions (with arbitrary covariance matrices) satisfy hypercontractivity, a practical example where hypercontractivity holds is the case when the state transition has Gaussian noise (as common in control settings). We will add more discussion in the next version.
>
>
> For the agnostic setting where realizability holds approximately, we believe a similar proof will work. However, in this case, the required agnostic error will depend on the gap. See [1] for more discussion on the agnostic setting.
>
> [1] Agnostic q-learning with function approximation in deterministic systems: Near-optimal bounds on approximation error and sample complexity.

---

> > ### Comment · Reviewer_PcYF · 2021-08-31
> > **Thanks, updated rating**
> >
> > I would like to thank the authors for addressing my questions and concerns. With this in mind, I have changed my rating from 7 to 8.

---

### Decision · Program_Chairs · 2021-09-27

**Decision:**

Accept (Oral)

**Comment:**

The reviewers all liked the paper very much. It will be a very nice addition to the conference program, presenting interesting and important additions to the growing body of literature on the theoretical analysis of RL algorithms with function approximation under the assumption that the action-value function of the optimal policy ($Q^*$) is linearly realizable.